# Semi-Supervised Preference Optimization with Limited Feedback

**Seonggyun Lee**[1], **Sungjun Lim**[1], **Seojin Park**[1], **Soeun Cheon**[2], **Kyungwoo Song**[1]*
[1]Yonsei University, [2]Korea Advanced Institute of Science and Technology
{leesg0104, lsj9862, juuck14, kyungwoo.song}@yonsei.ac.kr,
eintausend@kaist.ac.kr

## Abstract

The field of preference optimization has made outstanding contributions to the alignment of language models with human preferences. Despite these advancements, recent methods still rely heavily on substantial paired (labeled) feedback data, leading to substantial resource expenditures. To address these challenges, we study the problem of **S**emi-**S**upervised **P**reference **O**ptimization (**SSPO**) in which the idea is to learn from both a small number of pairwise preference labels and a large pool of unpaired samples simultaneously. Our key theoretical contribution proves the existence of an optimal reward threshold capable of separating winning and losing responses with high probability, which enables a principled pseudo-labeling of unpaired data. By leveraging these pseudo-labels, SSPO effectively distills latent preferences from large-scale unpaired data, thus maintaining human alignment while drastically reducing acquisition costs. Extensive experiments across datasets validate this remarkable data efficiency; for instance, SSPO trained with Mistral-7B-Instruct on just 1% of UltraFeedback consistently surpasses strong baselines trained on 10% of UltraFeedback.[1]

## 1 Introduction

Preference optimization (PO) is a pivotal method for aligning large language models (LLMs) with human values and expectations, ensuring that models provide useful, safe, and pleasant outputs (Calandriello et al., 2024). Without proper alignment, LLMs risk generating misleading or harmful content (Song et al., 2024). In PO, preferences reflect human judgments on the desirability or usefulness of responses, encompassing nuanced aspects such as ethical soundness or honesty (Christiano et al., 2017; Ouyang et al., 2022; Bai et al., 2022). Building such preference datasets typically involves labor-intensive human annotation processes (Cui et al., 2024; Karthik et al., 2024; Shi et al., 2024), where annotators craft prompts, gather responses from models (Achiam et al., 2023; Dubey et al., 2024), and rank them.

The primary bottleneck in PO is its profound reliance on data acquisition. This process is costly; it requires expert labor, averaging 5–10 minutes per comparison, and reaching $10–30 per data point (Bai et al., 2022; Casper et al., 2023). To conquer these limitations, recent work has explored synthetic feedback (Huang et al., 2023; Zhou et al., 2024; Xu et al., 2024) and automatic preference prediction using LLMs (He et al., 2024). While promising, these approaches face quality assurance issues due to the lack of verified ground truth.

A potential solution to this costly data acquisition lies in the vast amount of existing domain-specific data, such as question-answering pairs originally intended for supervised fine-tuning (SFT) (Agrawal et al., 2024). Although rich in expert knowledge, this data lacks the explicit preference labels required for PO. But it often contains valuable implicit preferences, such as coherent thinking patterns and appropriate stylistic tones. To leverage these unlabeled sources, some studies have attempted to generate preference signals using a capable LLM for self-annotation.

However, relying on synthetic data or LLM-based annotations for human alignment has fundamental limitations. This strategy risks creating a feedback loop in which an imperfectly aligned model

---

*Corresponding Author.

[1]https://github.com/MLAI-Yonsei/SSPO

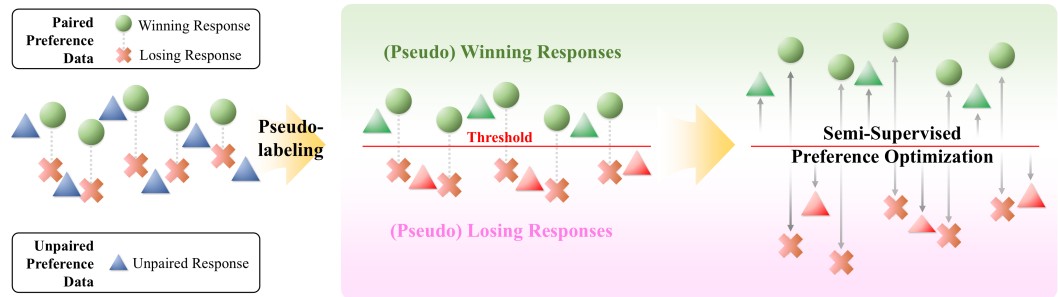

Figure 1: **Overview of the SSPO framework.** Existing preference optimization methods, such as DPO and SimPO, rely solely on a limited number of human-labeled comparisons. These methods discard abundant unpaired responses (e.g., supervised fine-tuning data) due to the lack of preference labels, which hinders generalization and data efficiency. SSPO leverages a reward function trained on labeled comparisons to assign pseudo-labels to unpaired responses. Responses above a learned threshold are treated as (pseudo) winning, and those below as (pseudo) losing. Hence, the policy model optimizes the reward threshold using both labeled and pseudo-labeled data, thereby improving alignment quality and generalization beyond the labeled dataset.

propagates its own biases and fails to fully capture the complexity of genuine human preferences, thus generating data of questionable reliability (Casper et al., 2023). Such synthetic preferences often lack the subtle, context-dependent nuances that are characteristic of true human judgment. Ultimately, acquiring reliable high-quality preference data, especially in specialized domains, still entails prohibitively expensive expert evaluation, creating a significant barrier to developing models that are knowledgeable and aligned with human values in a safe way (Miller et al., 2024).

To address this, we propose **S**emi-**S**upervised **P**reference **O**ptimization (SSPO), which combines a small amount of labeled preference data with large-scale SFT datasets via semi-supervised learning. SSPO reframes preference learning as a probabilistic classification task, enabling principled pseudo-labeling of unpaired data using a reward function trained on limited labeled examples. By adjusting a dynamic reward threshold, SSPO generates high-confidence preference labels and jointly optimizes the policy model with both paired and pseudo-labeled data. As shown in Figure 1, SSPO enables cost-efficient preference tuning while preserving domain knowledge and aligning with human values. It significantly reduces the reliance on large annotated datasets, offering a scalable and effective framework for high-quality LLM alignment.

## 2 RELATED WORK

### 2.1 PREFERENCE OPTIMIZATION

Preference optimization aligns language models with human preferences to improve helpfulness and safety. Proximal Policy Optimization (**PPO**) (Schulman et al., 2017) introduced a stable reinforcement learning algorithm that prevents large policy shifts during training. This method laid the foundation for preference-based fine-tuning of language models. Reinforcement Learning from Human Feedback (**RLHF**) (Ouyang et al., 2022) enhanced alignment by using human rankings to train a reward model, which then guides policy updates. While effective, RLHF's reliance on reward models can introduce bias and increase training complexity. To simplify this process, Rank Rewarding from Human Feedback (**RRHF**) (Yuan et al., 2023) and Direct Preference Optimization (**DPO**) (Rafailov et al., 2024) reduced dependence on reward models. RRHF directly incorporates ranking signals, while DPO frames preference learning as a binary classification task, improving stability and efficiency. Odds Ratio Preference Optimization (**ORPO**) (Hong et al., 2024) and Simple Preference Optimization (**SimPO**) (Meng et al., 2024) further improved efficiency by removing the need for a reference model. ORPO models preferences through odds ratios, while SimPO focuses on simplicity and ease of implementation. Kahneman-Tversky Optimization (**KTO**) (Ethayarajh et al., 2024) takes inspiration from behavioral economics, incorporating human-like biases (e.g., dislike of loss) into the optimization process to better reflect real-world preferences.

## 2.2 HUMAN ALIGNMENT WITH LIMITED FEEDBACK

Alignment with insufficient feedback is a key challenge in fine-tuning LLMs, where high-quality human preference data is often limited or costly (Casper et al., 2023). Ziegler et al. (2019) demonstrated the effectiveness of preference modeling for alignment but relied heavily on extensive human annotations. To reduce this dependency, Kim et al. (2025) and Huang et al. (2023) proposed generating synthetic preference data, and Zhou et al. (2024) extended this with a generalizable synthesis framework. Shi et al. (2024) and Liu et al. (2023) explored automated preference signals, while Semi-Supervised Reward Modeling **(SSRM)** (He et al., 2024) investigates reward modeling using a self-training framework under a semi-supervised setting.

## 2.3 SYNTHETIC PREFERENCE GENERATION

Synthetic preference generation reduces dependence on manual labeling but is intrinsically limited by the alignment quality of the generating models. For example, **AlpacaFarm** (Dubois et al., 2023) uses LLMs to simulate human preferences, improving cost efficiency but risking the propagation of model biases, which creates a circular constraint on alignment. Meanwhile, off-the-shelf supervised fine-tuning (SFT) data contains valuable implicit preference signals, such as reasoning and stylistic cues, yet is often underutilized. Self-training approaches leverage this data via pseudo-labeling (Wang et al., 2024), but typically require iterative annotation cycles that may introduce noise and instability. In comparison, Spread Preference Annotation **(SPA)** (Kim et al., 2025) addresses scarcity by repeatedly self-annotating with improved preference models to refine alignment progressively, though this process can be computationally intensive and may amplify errors.

## 3 PRELIMINARIES

In this section, we reformulate preference optimization as a Bayes-optimal classification problem. This reframing is a key to theoretically justifying our pseudo-labeling strategy for unpaired data in Section 4. We argue that aligning language model outputs with human preferences can be conceptualized as a probabilistic classification task over pairwise comparisons. The objective is to learn a scoring function that, given a prompt and two candidate responses, reliably identifies the preferred response. This perspective offers a principled theoretical framework for analyzing preference optimization.

We begin by modeling the preference classifier as a probabilistic binary function that assigns a higher score to the winning response. We then examine how the separation of reward values emerges after proper learning, allowing for a threshold-based pseudo-labeling strategy. Finally, we show that the reward-based ranking is consistent with model likelihoods under mild assumptions, providing a solid theoretical basis for our approach in Section 4.

Let $f_\theta : (x, y, y') \to [0, 1]$ denote a binary preference classifier parameterized by $\theta$ which takes a prompt $x$ and two candidate responses $y$ and $y'$ as input, and outputs a confidence score of human alignment of the model. Let $s \in \{0, 1\}$ be a true preference label, where $s = 1$ indicates that $y$ is preferred over $y'$, and $s = 0$ otherwise. We want to find the Bayes optimal classifier $f^*(x, y, y')$ $= \mathbb{P}(s \mid x, y, y') \propto \mathbb{P}(x, y, y' \mid s) \cdot \mathbb{P}(s)$ that minimizes the expected risk $R(f_\theta)$:

$$f^* = \arg\min_\theta R(f_\theta) \quad \text{where} \quad R(f_\theta) = \mathbb{E}[\ell(f_\theta, s)] \tag{1}$$

where the preference classifier is trained to predict the correct value of $s$ by minimizing the binary cross-entropy loss $\ell(f_\theta, s) = -s \log f_\theta(x, y, y') - (1 - s) \log(1 - f_\theta(x, y, y'))$.

To model our preference classifier, we define $f_\theta(x, y, y')$ by Bradley-Terry modelization (Bradley & Terry, 1952; Rafailov et al., 2024; Meng et al., 2024):

$$f_\theta(x, y, y') := \sigma(r_\theta(x, y) - r_\theta(x, y')) \cdot \mathbb{P}(s = 1) + \sigma(r_\theta(x, y') - r_\theta(x, y)) \cdot \mathbb{P}(s = 0), \tag{2}$$

where $r_\theta(x, y)$ is a reward function that reflects the likelihood that $y$ is preferred given $x$, and $\sigma(\cdot)$ is the sigmoid function. Each term in Eq. (2) corresponds to the confidence of the model in each possible preference direction. Namely, the first term represents the model's estimated probability that $y$ is preferred over $y'$ scaled by the prior probability that $s = 1$, and the second term accounts for the reverse case where $y'$ is preferred over $y$. This formulation models the expected prediction of the preference label $s$ by marginalizing over both possible outcomes.

Given that the human-annotated preference dataset $D_L = \{(x^{(i)}, y_w^{(i)}, y_l^{(i)})\}_{i=1}^{n_L}$ always contains pairs where $y_w^{(i)}$ is the winning response and $y_l^{(i)}$ is the losing one, we always have $(x, y, y') = (x, y_w, y_l)$, i.e., the preference label $s$ is always 1. This simplifies the risk function of the paired data $D_L$ as Eq. (3), which means that the risk function is equal to the objective of preference optimization:

$$
\begin{aligned}
\mathbb{E}_{D_L}[\ell(f_\theta, s)] &= \mathbb{E}_{D_L}[\ell(f_\theta, 1) \mid s = 1] \cdot \mathbb{P}_{D_L}(s = 1) + \mathbb{E}_{D_L}[\ell(f_\theta, 0) \mid s = 0] \cdot \mathbb{P}_{D_L}(s = 0) \\
&= \mathbb{E}_{D_L}[\ell(f_\theta, 1)] \\
&= \mathbb{E}_{D_L}[-\log \sigma(r_\theta(x, y_w) - r_\theta(x, y_l))].
\end{aligned}
\tag{3}
$$

This formulation captures the expected agreement of the classifier with the given preference label and serves as the foundation for reward-based preference learning. In our study, we set $r_\theta(x, y) = \frac{\beta}{|y|} \log \pi_\theta(y \mid x)$ as the default choice, which is the reward function of SimPO (Meng et al., 2024). Therefore, the expected risk of the paired data is:

$$
R_{D_L}(f_\theta) = \mathbb{E}_{D_L}\left[ -\log \sigma\left( \frac{\beta}{|y_w|} \log \pi_\theta(y_w \mid x) - \frac{\beta}{|y_l|} \log \pi_\theta(y_l \mid x) - \Delta \right) \right]
\tag{4}
$$

where $\pi_\theta(y \mid x)$ is the language model's policy (likelihood), $\beta$ is a scaling hyperparameter and $\Delta$ is a reward margin guarantee for winning to exceed losing (Firth, 2005; Meng et al., 2024).

# 4 PREFERENCE OPTIMIZATION WITH BOTH PAIRED AND UNPAIRED DATA

Acquiring paired preference data $(D_L)$ is often prohibitively expensive, making it challenging to collect a dataset sufficient for achieving desired performance levels. To overcome this, we leverage a large corpus of unpaired data, $D_U = \{(x_u^{(j)}, y_u^{(j)})\}_{j=1}^{n_U}$, which is more readily available such that $n_L \ll n_U$. Our core strategy involves setting a reward threshold using the paired data during training, applying it to assign pseudo-labels to the unpaired data, and adopting a curriculum that initially focuses on the paired data before gradually shifting toward the pseudo-labeled unpaired data. When trained on a small set of paired comparisons from $D_L$, $r_\theta$ learns to assign consistently higher values to winning responses than to losing ones. This creates a clear separation in the reward space, which enables the extension of preference supervision to the abundant unpaired data in $D_U$.

## 4.1 PSEUDO-LABELING WITH A REWARD THRESHOLD

Since our unpaired data consists of single prompt-response pairs, no preference comparison is available. We now introduce an *imaginary pair* by assuming a counterpart $y_b$ for each $y_u$ to form a virtual comparison triplet $(x_u, y_u, y_b)$. Here, $y_b$ represents a hypothetical response whose reward is precisely at the decision boundary we aim to learn. This formulation allows us to apply the same preference classifier $f_\theta$ defined in Section 3 to unpaired data. Then we can rewrite Eq. (2) with respect to the unpaired data as follows:

$$
\begin{aligned}
f_\theta(x_u, y_u, y_b) &= \sigma(r_\theta(x_u, y_u) - r_\theta(x_u, y_b)) \cdot \mathbb{P}_{D_U}(s = 1) \\
&+ \sigma(r_\theta(x_u, y_b) - r_\theta(x_u, y_u)) \cdot \mathbb{P}_{D_U}(s = 0).
\end{aligned}
\tag{5}
$$

However, the *imaginary* reward value $r_\theta(x_u, y_b)$ is not actually observable. To circumvent this, we propose a threshold-based strategy to infer a pseudo-label $\tilde{s} \in \{0, 1\}$ for each unpaired response $y_u$ based on its reward value $r_\theta(x_u, y_u)$. If the reward exceeds a certain threshold, we assign $\tilde{s} = 1$ (i.e., $y_u$ is likely to be a winning response); otherwise, we assign $\tilde{s} = 0$.

A key question, then, is how to define this threshold in a principled and reliable way. Let $r$ be the reward score for response $y$ on input $x$, and let $\delta$ be a reward threshold. We define the *Bayes risk* of $\delta$ as the total misclassification probability when using $\delta$ as a hard decision boundary. Minimizing this risk is crucial as it provides the most statistically robust method for finding a threshold that minimizes the chances of incorrectly labeling a losing response as winning, or vice versa. The risk is formally defined as:

$$
R(\delta) = \mathbb{P}(s = 1) \cdot \int_{-\infty}^{\delta} p(r \mid s = 1) \, dr + \mathbb{P}(s = 0) \cdot \int_{\delta}^{\infty} p(r \mid s = 0) \, dr,
\tag{6}
$$

where $p(r \mid s)$ is the density of reward scores conditioned on true preference label $s$. This expression reflects the area of overlap between the reward distributions of winning and losing responses. We define the *optimal threshold* $\delta^*$ as the one that minimizes this misclassification risk:

$$\delta^* = \arg\min_{\delta \in \mathbb{R}} R(\delta). \tag{7}$$

With this definition in place, we can now state Theorem 1 to guarantee the existence of such an optimal $\delta^*$. We propose to replace the unobservable $r_\theta(x_u, y_b)$ with a threshold $\delta$, such that comparisons against $\delta$ reflect the preference tendencies learned from the paired data. We first show that, under mild distributional assumptions, such a threshold exists with high probability. This threshold reliably separates the reward values of winning and losing responses, enabling us to perform pseudo-labeling in a theoretically sound manner. The proof is provided in Appendix B.1.

**Theorem 1.** *(Existence of an Optimal Reward Threshold) Let us consider the i.i.d. samples of rewards from losing responses, $\{r_\theta(x^{(i)}, y_l^{(i)})\}_{i=1}^{n_L}$, and from winning responses, $\{r_\theta(x^{(j)}, y_w^{(j)})\}_{j=1}^{n_L}$. Assume both distributions are sub-Gaussian with means $\mu_l, \mu_w$ and variance proxies $\sigma_l^2, \sigma_w^2$, which formulates $\mu_w > \mu_l$. Then, for any $\alpha \in (0,1)$ and non-negative $t_1$ and $t_2$ satisfying $n_L \cdot (\exp(-t_1^2/2\sigma_l^2) + \exp(-t_2^2/2\sigma_w^2)) \leq \alpha$, there exists an optimal reward threshold $\delta^* = \mu_l + t_1 = \mu_w - t_2$ such that*

$$\mathbb{P}\left(\max_i r_\theta(x^{(i)}, y_l^{(i)}) \leq \delta^* \leq \min_j r_\theta(x^{(j)}, y_w^{(j)})\right) \geq 1 - \alpha \tag{8}$$

*for all $i, j \in \mathcal{I}(D_L)$, where $\mathcal{I}(D_L)$ denotes the index set of instances in the paired data $D_L$.*

Here, the expression $\delta^* = \mu_l + t_1 = \mu_w - t_2$ should be understood as selecting a representative threshold inside the high-probability interval $[\mu_l + t_1, \mu_w - t_2]$ rather than as a strict equality that must hold for arbitrary reward distributions. In practice, any value in this interval yields the same separation guarantee, and it simply asserts that such an interval exists under the stated sub-Gaussian assumptions. Therefore, Theorem 1 guarantees the existence of at least one threshold within the interval $[\mu_l + t_1, \mu_w - t_2]$ that separates the two sets of rewards with probability at least $1 - \alpha$, thereby underpinning the conceptual soundness of threshold-based pseudo-labeling. Therefore, this serves as a conceptual foundation for our framework; it confirms that in a regime where the preference model has learned to distinguish responses with a sufficient margin, a threshold-based decision boundary provides a statistically sound strategy for pseudo-labeling.

## 4.2 PRACTICAL APPROACH

While Eq. (8) guarantees that an optimal threshold $\delta^*$ exists within a certain range, it depends on the unknown means $\mu_l$ and $\mu_w$. Therefore, we cannot compute $\delta^*$ directly in practice. To address this limitation, we estimate the reward densities of winning and losing responses in $D_L$ via kernel density estimation (Parzen, 1962; Silverman, 2018) and solve for the threshold directly:

$$\hat{p}_w(r) = \frac{1}{n_L \cdot h} \sum_{j=1}^{n_L} \mathcal{K}\left(\frac{r - r_\theta(x^{(j)}, y_w^{(j)})}{h}\right), \quad \hat{p}_l(r) = \frac{1}{n_L \cdot h} \sum_{i=1}^{n_L} \mathcal{K}\left(\frac{r - r_\theta(x^{(i)}, y_l^{(i)})}{h}\right), \tag{9}$$

where $\mathcal{K}(u) = \frac{1}{\sqrt{2\pi}} \exp(-\frac{1}{2}u^2)$ is the Gaussian kernel and $h$ is the bandwidth. We search numerically for the threshold $\hat{\delta}$ that minimizes the estimated Bayes risk $\hat{R}(\delta)$:

$$\hat{\delta} = \arg\min_{\delta \in \mathbb{R}} \hat{R}(\delta), \quad \text{where} \quad \hat{R}(\delta) = \mathbb{P}(s = 1) \cdot \int_{-\infty}^{\delta} \hat{p}_w(r)\, dr + \mathbb{P}(s = 0) \cdot \int_{\delta}^{\infty} \hat{p}_l(r)\, dr. \tag{10}$$

This practical threshold $\hat{\delta}$ is used to assign pseudo-labels to the unpaired responses in $D_U$, thereby serving as a principled boundary that separates winning from losing responses based on their reward scores. Thus, we reinterpret Eq. (5) with the practical threshold $\hat{\delta}$ to define the pseudo-labeled risk over dataset $D_U$ as follows:

$$R_{D_U}(f_\theta) = \frac{1}{n_U} \sum_{k=1}^{n_U} \ell(f_\theta, \tilde{s}_k) \cdot \mathbb{P}_{D_U}(s = \tilde{s}_k) \quad \text{with} \quad \tilde{s}_k = \mathbb{I}\left\{r_\theta(x_u^{(k)}, y_u^{(k)}) > \hat{\delta}\right\}, \tag{11}$$

where $\mathbb{P}_{D_U}(s = 1)$ is a prior probability that the unpaired response is preferred over its hypothetical counterpart. This prior captures our initial belief about the proportion of winning responses in the unpaired data and is fixed throughout training. Eq. (11) bridges the gap between paired and unpaired data by treating confident unpaired responses as if they were weakly labeled. Unlike heuristic filtering, our approach grounds pseudo-labeling in a statistically principled threshold derived from the reward distribution. Building on this formulation, we now introduce an adaptive scheduling strategy to effectively balance the influence of paired and unpaired data during optimization. For detailed implementation of the pseudo-labeled risk, see Appendix C.1.

### 4.3 ADAPTIVE SCHEDULING FOR CURRICULUM LEARNING

To effectively combine paired ($D_L$) and unpaired ($D_U$) data, we introduce an *adaptive scheduler* instead of using a fixed weighting factor. This scheduler implements a curriculum learning strategy by dynamically adjusting the coefficient $\gamma'$. Initially, the model prioritizes the more reliable supervision from paired data. As training progresses and the reward function improves, the weight of the unpaired data gradually increases. Our final training objective is defined in Eq. (12):

$$\mathcal{L}(f_\theta) = \gamma' \cdot R_{D_L}(f_\theta) + (1 - \gamma') \cdot R_{D_U}(f_\theta) \quad \text{s.t.} \quad \gamma' = \max\left\{\gamma_{\min}, \gamma_0 \cdot \exp(-\lambda\tau)\right\}, \quad (12)$$

where $\gamma' \in (0, 1]$ is the adaptive coefficient at training step $\tau$ ($1 \leq \tau \leq \mathcal{T}$), $\lambda > 0$ is the decay rate, and $\gamma_0$ is the initial value. We set $\gamma_0 = 1$ to focus on paired data at the start and define the minimum value as the proportion of paired data, i.e., $\gamma_{\min} = n_L/(n_L + n_U)$. We provide a gradient analysis and the complete algorithm in Appendix B.2 and C.4, respectively.

## 5 EXPERIMENTS

### 5.1 TOY EXPERIMENT

**Setup.** To analyze SSPO's behavior, we design two synthetic experiments: one noise-free and one with label noise. These experiments serve as a sanity check and provide interpretable insights before applying SSPO to real-world preference datasets. Each prompt consists of ten words randomly sampled from the NLTK words (Loper & Bird, 2002), with the shortest word designated as preferred over the longest. We fix the unpaired dataset size at 1,000 samples and vary the paired dataset size (10, 50, 100) to simulate data-scarce scenarios. Using **GPT-2-small** (Radford et al., 2019) as a reward model, we compare SSPO against strong baselines under various prior values (0.1, 0.3, 0.5, 0.7, 0.9). To evaluate robustness, the noisy setting *swaps* the designations of winning and losing for 10%, 30%, and 50% of the paired preference labels.

Table 1: **Comparison of test accuracy on toy dataset without or with noise in paired data.** SSPO consistently and significantly outperforms all baselines across different quantities of paired data.

| Method | $n_L = 10$ | $n_L = 50$ | $n_L = 100$ |
|---|---|---|---|
| *Noise 0%* | | | |
| DPO | 0.743 | 0.777 | 0.846 |
| ORPO | 0.590 | 0.679 | 0.710 |
| SimPO | 0.762 | 0.776 | 0.817 |
| **SSPO** (0.5) | **0.841** | **0.879** | **0.960** |
| *Noise 50%* | | | |
| DPO | 0.571 | 0.567 | 0.554 |
| ORPO | 0.594 | 0.586 | 0.549 |
| SimPO | 0.594 | 0.563 | 0.534 |
| **SSPO** (0.5) | **0.757** | **0.656** | **0.563** |

**Results.** Table 1 shows the part of the experimental results in the noise-free or noisy settings, where SSPO consistently outperforms all baseline methods across different quantities of paired data. The performance gap is especially notable in data-scarce scenarios, highlighting SSPO's effectiveness in leveraging unpaired preference data even with limited labeled pairs. Under a noisy condition, SSPO also maintains superior performance, which demonstrates its robustness to label noise and supports its practical value in real-world settings where clean preference annotations are costly or difficult to obtain. Full experimental results and detailed settings are provided in Appendix C.2.

### 5.2 REAL-DATA EXPERIMENTS

**Baselines.** To evaluate the effectiveness of SSPO in realistic settings, we compare it against a comprehensive set of baseline methods. For preference optimization under limited supervision, we consider **DPO**, **ORPO**, **SimPO**, and **KTO**. Additionally, we compare against **SSRM** and **SPA**, which are specifically designed to reduce reliance on large-scale preference data through iterative pseudo-labeling, or iterative generation of self-annotated preference data.

Table 2: **Performance of AlpacaEval2.0(%) and MT-Bench.** LC and WR denote length-controlled and raw win rates for AlpacaEval2.0, and MT is the average MT-Bench score. With just 1% of paired data, SSPO often achieves higher scores than baselines trained on 10% of the data, exhibiting its data efficiency and effectiveness. The best numbers are in bold, and the second-best ones are underlined.

| Baseline | Size | UltraFeedback | | | | | | | | |
| | | Phi-2 (2.7B) | | | Mistral (7B) | | | Llama3 (8B) | | |
| | | LC | WR | MT | LC | WR | MT | LC | WR | MT |
|---|---|---|---|---|---|---|---|---|---|---|
| DPO | 1% | 3.6 | 2.5 | **6.3** | 17.0 | 12.8 | 7.6 | 12.1 | 12.6 | **8.0** |
| | 10% | 4.6 | 2.6 | **6.3** | 18.0 | 13.6 | 7.6 | 13.0 | 13.7 | 8.0 |
| ORPO | 1% | 3.7 | 2.3 | **6.3** | 15.0 | 11.4 | 7.5 | 9.4 | 10.3 | **8.0** |
| | 10% | 3.9 | 2.6 | **6.3** | 16.7 | 10.9 | 7.5 | 10.0 | 10.9 | 7.9 |
| SimPO | 1% | 4.0 | 2.5 | **6.3** | 13.2 | 8.3 | 7.6 | 14.3 | 15.1 | **8.0** |
| | 10% | 4.0 | 2.5 | **6.3** | 18.1 | 12.9 | 7.5 | 13.0 | 13.7 | 7.9 |
| SSRM | 1% | 4.3 | 2.0 | 6.2 | 14.9 | 13.2 | 5.4 | **14.9** | 12.8 | 6.2 |
| | 10% | 4.4 | 2.6 | **6.3** | 16.2 | 13.3 | 5.5 | 15.1 | 16.3 | 6.2 |
| KTO | 1% | 4.0 | 2.5 | **6.3** | 16.4 | 14.9 | 7.6 | 14.4 | 15.8 | 7.9 |
| | 10% | 4.4 | 2.6 | **6.3** | 18.8 | 16.4 | 7.6 | 16.7 | 18.2 | 8.0 |
| SPA | 1% | 4.0 | 2.6 | **6.3** | 18.2 | 15.6 | **7.7** | 13.4 | 15.3 | 7.9 |
| | 10% | 4.9 | 3.1 | **6.3** | 19.1 | 18.7 | **7.8** | 14.5 | 16.6 | **8.1** |
| **SSPO** | 1% | **7.2** | **4.1** | **6.3** | **26.7** | **18.1** | **7.7** | 14.8 | **16.0** | **8.0** |
| | 10% | **7.7** | **4.3** | **6.3** | **30.0** | **20.7** | 7.7 | **20.7** | **20.8** | 7.9 |

| Baseline | Size | UltraMedical-Preference | | | | | | DSP Business | | | | | |
| | | Mistral (7B) | | | Llama3 (8B) | | | Mistral (7B) | | | Llama3 (8B) | | |
| | | LC | WR | MT | LC | WR | MT | LC | WR | MT | LC | WR | MT |
|---|---|---|---|---|---|---|---|---|---|---|---|---|---|
| DPO | 1% | **8.7** | 4.8 | 5.2 | 2.6 | 5.3 | 6.5 | 15.0 | 6.5 | 6.7 | 2.7 | 2.1 | **5.6** |
| | 10% | 11.2 | 7.2 | **5.3** | 7.8 | 6.2 | 6.4 | 16.0 | 6.8 | 6.8 | 3.7 | 3.6 | 5.7 |
| ORPO | 1% | 3.7 | 3.1 | 5.2 | 2.2 | 4.8 | 6.4 | 15.9 | 6.5 | 6.8 | 3.6 | 3.1 | **5.6** |
| | 10% | 6.5 | 5.0 | 5.2 | 7.5 | 5.6 | 6.4 | 16.0 | 7.0 | 6.8 | 4.6 | 4.7 | 5.6 |
| SimPO | 1% | 6.0 | 7.2 | **5.3** | 2.8 | 3.8 | 6.4 | 15.5 | 6.5 | 6.7 | 2.7 | 2.1 | **5.6** |
| | 10% | 10.2 | 7.6 | **5.3** | 10.5 | 4.5 | 6.5 | 15.9 | 7.1 | 6.8 | 3.6 | 3.5 | 5.7 |
| SSRM | 1% | 6.0 | 5.9 | 5.2 | 3.9 | 5.0 | 6.4 | 15.2 | 6.7 | 6.6 | 3.1 | 2.9 | **5.6** |
| | 10% | 13.1 | 14.6 | **5.3** | 12.9 | 15.8 | 6.4 | 15.9 | 6.9 | 6.8 | 3.8 | 3.5 | 5.6 |
| KTO | 1% | 6.4 | 9.8 | 5.2 | 3.7 | 5.4 | 6.4 | 15.7 | 6.8 | 6.6 | 3.2 | 3.1 | **5.6** |
| | 10% | 11.1 | 15.6 | **5.3** | 14.2 | 15.2 | 6.4 | 16.7 | 7.5 | 6.8 | 4.4 | 4.6 | 5.7 |
| SPA | 1% | 6.7 | 10.1 | 5.2 | 4.3 | 5.8 | 6.4 | 15.4 | 6.5 | 6.7 | 3.0 | 2.7 | **5.6** |
| | 10% | 11.3 | 16.2 | **5.3** | 15.7 | 16.9 | 6.4 | 16.0 | 7.2 | 6.9 | 4.3 | 4.1 | 5.7 |
| **SSPO** | 1% | 7.6 | **13.2** | 5.2 | **5.1** | **6.7** | **6.7** | **17.2** | **7.1** | **6.9** | **3.7** | **3.6** | **5.6** |
| | 10% | **12.0** | **17.0** | **5.3** | **17.7** | **18.4** | **6.9** | **17.9** | **8.8** | **7.0** | **5.7** | **5.7** | 5.6 |

**Training.** We conduct training using **UltraFeedback** (Cui et al., 2024) for paired preference data. To simulate data-scarce scenarios, we evaluate on two challenging settings: using only 1% ($n_L$=611) or 10% ($n_L$=6,113) of each original dataset, including validation data. For SSPO, we use 10% of **UltraChat-200k** (Ding et al., 2023) ($n_U$=20,786) as the unpaired data to be pseudo-labeled. We utilize off-the-shelf models already fine-tuned on the UltraChat dataset as initial models. We employ **Phi-2 (2.7B)** (Javaheripi et al., 2023), **Mistral-7B-Instruct-v0.2** (Jiang et al., 2023), and **Llama3-8B-Instruct** (Dubey et al., 2024). to demonstrate the generality and scalability of SSPO.

We also conduct two domain-specific experiments to investigate the alignment with both domain expertise and human preferences. For the **medical** domain, we set two data sizes according to

UltraFeedback; we set 1% ($n_L$=1,093) and 10% ($n_L$=10,935) of **UltraMedical-Preference** including validation data, which are composed of pairwise comparisons constructed from real-world medical questions and multiple answers. For SSPO, we use 5% of **UltraMedical** (Zhang et al., 2024) ($n_U$=20,479) as the unpaired data to match the amount of data in accordance with UltraChat. We train **Meerkat-7B-v1.0** (Kim et al., 2024) and **Llama-3-8B-UltraMedical** (Zhang et al., 2024) on these datasets. Similarly, for the **business** domain, we set 1% ($n_L$=125) and 10% ($n_L$=1,255) of the **DSP Business** (Cheng et al., 2023), which provides business-oriented pairwise comparisons. For SSPO, we use whole **17k Business Book** ($n_U$=17,480) as the unpaired data. We train **Mistral-7B-Business** and **Finance-Llama-8B** on these datasets. For more implementation details and hyperparameter settings, please refer to Appendix C.3.

**Evaluation.** We adopt **AlpacaEval2.0** (Dubois et al., 2025) and **MT-Bench** (Zheng et al., 2023), two widely used benchmarks for assessing models' conversational and instruction-following abilities across diverse topics and domains. We use GPT-4-Turbo (Achiam et al., 2023) as the judge model and report both a raw win rate (WR) and a length-controlled win rate (LC), where the LC is the variant of WR explicitly controlling the response-length differences. This correction is designed to address the well-established verbosity bias in LLM-based evaluators, which often exhibit a systematic tendency to favor excessively long outputs irrespective of substantive quality. By conditioning on equal-length responses, LC yields preference estimates that more faithfully reflect semantic adequacy and instruction compliance. Meanwhile, MT-Bench is made up of 80 questions from 8 categories, and we investigate the average score of two turns on a scale of 10 using GPT-4 (Achiam et al., 2023) as the judge model. MT-Bench provides a complementary perspective by evaluating multi-turn dialog coherence, instruction adherence, and reasoning depth under carefully curated human-written prompts, making it a strong indicator of real-world conversational robustness.

## 5.3 MAIN RESULTS AND ABLATION STUDIES

**SSPO consistently and significantly outperforms all baseline methods across all backbones, data sizes, and domains.** As shown in Table 2, SSPO shows remarkable data efficiency, particularly in the most challenging, data-scarce settings. For instance, SSPO trained with Mistral on only 1% of UltraFeedback accomplishes a length-controlled win rate (LC) of 26.7%, surpassing the best-performing baseline (18.2% at SPA) in the same 1% setting and even all baselines trained with 10% of the data (e.g., 19.1% at SPA). In addition, on domain-specific datasets, i.e., UltraMedical-Preference and DSP Business, the LC of SSPO trained with Llama3-based models beat the strongest baseline regardless of data sizes. Therefore, the results validate that SSPO is a versatile and highly efficient preference optimization algorithm that effectively amplifies the preference signal through the principled use of abundant unpaired data. For more quantitative analyses, please refer to Appendix F.

**Sensitivity Analysis.** The assumed prior for unpaired data, $\mathbb{P}_{D_U}(s = 1)$, is a key hyperparameter that scales the influence of pseudo-labeled samples. Our ablation study, presented in Table 3, reveals that SSPO is robust to the choice of prior and a balanced prior of 0.5 consistently yields the best performance across all models. Deviating from this neutral stance (e.g., priors of 0.1 or 0.9) degrades performance, as overly confident priors can cause the model to overfit to noisy pseudo-labels. Critically, however, even with the most suboptimal priors, SSPO's performance remains strong, substantially outperforming the baselines in Table 2. This highlights the overall durability of our method and confirms that a prior of 0.5 is a safe and effective default choice.

Table 3: **SSPO performance when varying the assumed prior.** We measure the LC and WR for models trained on 10% of UltraFeedback. SSPO remains robust even under suboptimal priors, consistently outperforming baselines.

| Prior | Phi-2 | | | Mistral | | | Llama3 | | |
|---|---|---|---|---|---|---|---|---|---|
| | LC | WR | MT | LC | WR | MT | LC | WR | MT |
| 0.1 | 6.8 | 3.8 | 6.0 | 25.6 | 17.5 | 7.5 | 17.8 | 18.4 | 7.7 |
| 0.3 | 7.3 | 4.2 | 6.1 | 29.0 | 19.0 | 7.6 | 19.9 | 20.0 | 7.8 |
| **0.5** | **7.7** | **4.3** | **6.3** | **30.0** | **20.7** | **7.7** | **20.7** | **20.8** | **7.9** |
| 0.7 | 7.6 | **4.3** | 6.2 | 28.8 | 18.5 | 7.6 | **20.7** | 20.1 | 7.8 |
| 0.9 | 7.4 | 4.1 | 6.2 | 25.7 | 17.6 | 7.5 | 19.5 | 19.6 | 7.5 |

**Ablation on Adaptive Scheduling.** To verify the importance of our adaptive scheduler from Eq. (12), we conduct an ablation study comparing the standard SSPO against variants that use a fixed $\gamma'$ throughout training. Table 4 shows that the adaptive scheduler plays a vital role in harnessing both reliable and uncertain data sources, while SSPO with a fixed $\gamma'$ value still demonstrates strong performance compared to the baselines. We visualize the core mechanism behind the scheduler's

success in Figure 2. We define the loss contribution ratio at training step $\tau$ as $\gamma' \cdot R_{D_L}(f_\theta)/(\gamma' \cdot R_{D_L}(f_\theta) + (1 - \gamma') \cdot R_{D_U}(f_\theta))$. As the total training loss successfully converges (top plot), the bottom plot reveals a clear and principled transition by showing the contribution ratio of the paired loss. This illustrates how the adaptive scheduler dynamically shifts learning focus; it starts with the reliable paired loss from UltraFeedback ($R_{D_L}(f_\theta)$, cyan) and, as training stabilizes, increasingly relies on pseudo-labeled unpaired data from UltraChat ($R_{D_U}(f_\theta)$, red), which ultimately dominates. This transition highlights that adaptive scheduling is crucial for robustly leveraging both trusted and uncertain data sources to achieve superior alignment.

Table 4: **SSPO Performance with or without the adaptive scheduler.** ✓ denotes the case with adaptive scheduling, while ✗ indicates the case without it. The adaptive scheduler unlocks the method's full potential and consistently achieves stronger performance than baselines, even when the $\gamma'$ is fixed.

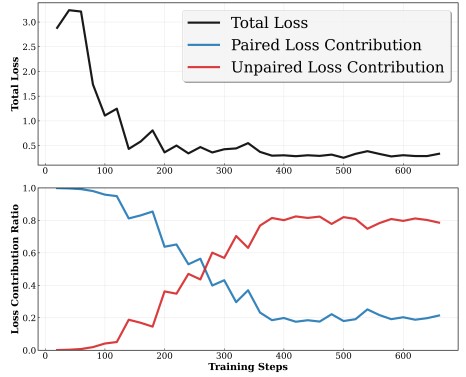

| Size | Sched. | Phi-2 | | | Mistral | | | Llama3 | | |
|---|---|---|---|---|---|---|---|---|---|---|
| | | LC | WR | MT | LC | WR | MT | LC | WR | MT |
| 1% | ✓ | 7.2 | 4.1 | 6.3 | 26.7 | 18.1 | 7.7 | 15.0 | 16.1 | 8.0 |
| | ✗ | | | | | | | | | |
| | $\gamma'$=0.1 | 6.5 | 3.9 | 6.2 | 24.1 | 16.5 | 7.5 | 13.2 | 14.8 | 7.9 |
| | $\gamma'$=0.5 | 6.8 | 3.8 | 6.3 | 26.0 | 17.2 | 7.6 | 14.1 | 15.5 | 7.9 |
| 10% | ✓ | 7.7 | 4.3 | 6.3 | 30.0 | 20.7 | 7.7 | 20.7 | 20.8 | 7.9 |
| | ✗ | | | | | | | | | |
| | $\gamma'$=0.1 | 7.2 | 4.0 | 6.3 | 27.5 | 18.1 | 7.6 | 18.5 | 19.1 | 7.9 |
| | $\gamma'$=0.5 | 7.3 | 4.2 | 6.3 | 29.3 | 19.8 | 7.7 | 19.6 | 20.0 | 7.9 |

Figure 2: **Loss Contribution Ratio.** (Mistral trained on 1% of UltraFeedback) This illustrates how the adaptive scheduler shifts the model's learning focus from paired data (cyan) to pseudo-labeled unpaired data (red), enabling effective and robust learning.

**Robustness against Confirmation Bias and Non-stationary Rewards.** To address the risks of confirmation bias and reward non-stationarity, we analyze the evolution of reward distributions and the estimated threshold in Figure 3. In the early stages (e.g., step 100), where winning and losing distributions exhibit significant overlap, the adaptive scheduler $\gamma'$ prioritizes the supervised loss from $D_L$. This acts as a reliable anchor to prevent noise amplification from incorrect pseudo-labels, aligning with the *early learning* phenomenon where models learn clean patterns before overfitting to noise (Arazo et al., 2020). As training progresses and a clear margin emerges, SSPO addresses the shifting nature of reward values by estimating the optimal boundary via KDE and stabilizing it with EMA. Drawing on consistency regularization principles (Tarvainen & Valpola, 2018), this dynamic thresholding ensures the decision boundary $\hat{\delta}$ consistently tracks the optimal separation without collapsing, enabling the model to effectively leverage unpaired data $D_U$ for generalization.

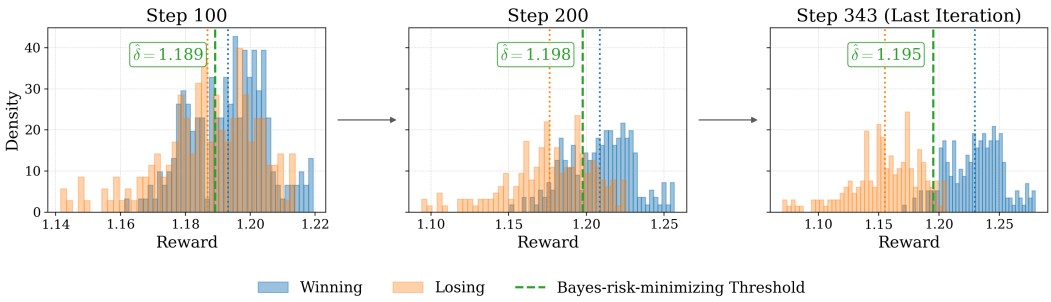

Figure 3: **Evolution of reward distributions and the Bayes-risk-minimizing threshold during SSPO training.** We visualize the reward densities of winning (blue) and losing (orange) responses generated by each training step of Mistral trained with 10% of the DSP Business. The dashed green line indicates the estimated threshold $\hat{\delta}$. As training progresses, the separation between the two distributions becomes distinct, and the adaptive threshold tracks the optimal decision boundary.

Table 5: **A case study showing how SSPO benefits from semantically relevant but unpaired data** (Mistral trained on 10% of **UltraFeedback**). SSPO's answer adopts a detailed, well-structured format (e.g., list of materials, step-by-step instructions) observed in the high-quality unpaired response, leading to a more helpful and user-friendly guide than our baselines. For brevity, some contents are omitted.

| Item | Content |
| --- | --- |
| **Question** | Can you help me figure out how to sew a button on a shirt? |
| **KTO** | You can just take a button and a shirt and then sew it on. You just need to attach it with some thread. |
| **Pseudo-Winning Datum** (UltraChat) | **Instruction:** Please provide step-by-step instructions on how to sew a button onto a shirt, including the type of thread to use, the needle size, and the specific sewing technique required. Additionally, please include tips for ensuring the button is securely attached and evenly aligned on the shirt. 
 **Response:** Materials Needed: Button, Shirt, Thread (preferably a heavy-duty or button/crafting thread), Needle (preferably a sharp, larger needle like a size 16) *(omitted)* Instructions: 1. Choose your button *(omitted)* 9. After you've sewn the button on securely, end by bringing the needle through the fabric to the inside of the shirt and knotting the thread. *(omitted)* Tips: *(omitted)* |
| **SSPO** (Ours) | Absolutely! Here's a step-by-step guide on how to sew a button onto a shirt: 1. Gather your supplies: You will need a button, thread that matches the color of your shirt, a sewing needle, a thimble (optional), and a pair of scissors. 2. Thread your needle: Thread the needle with a length of thread that is about 18 inches long. *(omitted)* 6. Trim the thread: Use the scissors to cut off any excess thread. Congratulations! You have successfully sewn a button onto a shirt. If you found this guide helpful, please let me know and feel free to ask if you have any other questions. |

### 5.4 Latent Preference in the Unpaired Data: Case Study

The effectiveness of SSPO stems from its capability to uncover and leverage latent preference signals hidden within unpaired data. While traditional preference optimization methods discard unpaired responses due to the lack of explicit preference labels, SSPO demonstrates that these responses contain valuable implicit preference information that can be extracted through principled pseudo-labeling.

**Substring Matching.** To identify the specific benefits of unpaired data in SSPO, we conducted a comprehensive analysis using a substring matching algorithm to find semantically similar content between AlpacaEval2.0 questions and our unpaired datasets. For each AlpacaEval2.0 question, we computed semantic similarity scores with both paired and unpaired data, identifying cases where questions had high similarity to unpaired content ($\geq 0.6$) but low similarity to paired data ($\leq 0.3$). This analysis revealed numerous instances where SSPO's performance was enhanced by exposure to relevant unpaired data during training. Table 5 presents a concrete example where SSPO benefited from unpaired data exposure in the UltraChat. We show the AlpacaEval2.0 question, the most similar instruction and corresponding response in each unpaired data, and the answer to the question from fine-tuned models. SSPO successfully aligns its generation based on such high-quality unpaired responses, resulting in better evaluation outcomes. More case studies and qualitative examples are provided in Appendix G.1 and G.2, respectively.

## 6 Conclusion

This work introduces SSPO, a theoretically grounded and practically effective framework for semi-supervised preference optimization. By reformulating preference alignment as a binary classification problem, we show that a reward function acts as a preference classifier. This formulation allows for a pseudo-labeling strategy that is theoretically justified by the separation of reward distributions. For practical application, we use a Bayes-risk-minimizing threshold derived from paired data and an adaptive scheduler that creates a curriculum learning dynamic, shifting the training focus from labeled to unpaired signals over time. Comprehensive experiments corroborate that SSPO is significantly superior to existing methods when paired data is scarce, while also scaling effectively with more unpaired data. Our findings open a promising direction for scalable alignment of language models with semi-supervised learning, without the need for large-scale human preference annotations.

ACKNOWLEDGEMENT

This work was supported by the National Research Foundation of Korea(NRF) grant funded by the Korea government(MSIT)(RS-2024-00457216).

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

## A    STATEMENT ON THE USE OF LARGE LANGUAGE MODELS

During the research, Large Language Models (LLMs) were utilized in a limited, supporting capacity. Their utilization was confined to post-drafting revisions, particularly for grammatical corrections, sentence refinement, and enhancing overall writing consistency. The core intellectual contributions, including the research ideas, methodology, data analysis, and conclusion, are solely the work of the authors. No substantive or analytical content was generated by LLMs, ensuring that their application did not compromise the academic integrity and originality of this research.

## B    THEORETICAL ANALYSIS

### B.1    PROOF OF THEOREM 1

We denote $r_l^{(i)} := r_\theta(x^{(i)}, y_l^{(i)})$ and $r_w^{(j)} := r_\theta(x^{(j)}, y_w^{(j)})$ for simplicity. Assume that $\{r_l^{(i)}\}_{i=1}^{n_L}$ and $\{r_w^{(j)}\}_{j=1}^{n_L}$ are i.i.d. samples from the reward distributions $\mathcal{F}_l$ and $\mathcal{F}_w$ respectively, which are sub-Gaussian with means $\mu_l, \mu_w$ and variance proxies $\sigma_l^2, \sigma_w^2$.

By the sub-Gaussian tail bound, for all $i, j \in \mathcal{I}(D_L)$, non-negative $t_1$ and $t_2$, we have:

$$\mathbb{P}\left(r_l^{(i)} \geq \mu_l + t_1\right) \leq \exp\left(-\frac{t_1^2}{2\sigma_l^2}\right), \quad \mathbb{P}\left(r_w^{(j)} \leq \mu_w - t_2\right) \leq \exp\left(-\frac{t_2^2}{2\sigma_w^2}\right).$$

We get union bounds respectively:

$$\mathbb{P}\left(\max_i r_l^{(i)} \geq \mu_l + t_1\right) \leq \sum_{i=1}^{n_L} \exp\left(-\frac{t_1^2}{2\sigma_l^2}\right) = n_L \cdot \exp\left(-\frac{t_1^2}{2\sigma_l^2}\right),$$

$$\mathbb{P}\left(\min_j r_w^{(j)} \leq \mu_w - t_2\right) \leq \sum_{j=1}^{n_L} \exp\left(-\frac{t_2^2}{2\sigma_w^2}\right) = n_L \cdot \exp\left(-\frac{t_2^2}{2\sigma_w^2}\right).$$

Then, the probability that all rewards of losing responses are below $\mu_l + t_1$ and all rewards of winning responses are above $\mu_w - t_2$ is at least

$$\mathbb{P}\left(\max_i r_l^{(i)} \leq \mu_l + t_1 \text{ and } \min_j r_w^{(j)} \geq \mu_w - t_2\right) \geq 1 - n_L \cdot \left(\exp\left(-\frac{t_1^2}{2\sigma_l^2}\right) + \exp\left(-\frac{t_2^2}{2\sigma_w^2}\right)\right).$$

Therefore, if any $\alpha \in (0, 1)$, $t_1$ and $t_2$ satisfy $\alpha \geq n_L \cdot (\exp(-t_1^2/2\sigma_l^2) + \exp(-t_2^2/2\sigma_w^2))$, the optimal reward threshold $\delta^* = \mu_l + t_1 = \mu_w - t_2$ lies between the maximum reward of losing responses and the minimum reward of winning responses with probability at least $1 - \alpha$. This completes the proof.

### B.2    GRADIENT ANALYSIS

We provide a detailed analysis of the gradient behavior of our objective, separately for the supervised (paired) and semi-supervised (unpaired) components. This analysis reveals how the model encourages correct ranking through reward-based learning signals, and how pseudo-labels influence parameter updates.

**Gradient on the paired data.**    We define our paired-data risk as Eq. (4). The gradient weight increases when the model incorrectly assigns a higher reward to the losing response. This encourages the model to correct such misrankings by increasing the winning score and decreasing the losing one. This analysis is basically the same as DPO (Rafailov et al., 2024) and SimPO (Meng et al., 2024).

**Gradient on the unpaired data.**    For unpaired data, each sample $(x_u^{(k)}, y_u^{(k)})$ is pseudo-labeled based on a threshold:

$$\tilde{s}_k = \begin{cases} 1, & r_\theta(x_u^{(k)}, y_u^{(k)}) > \hat{\delta}, \\ 0, & r_\theta(x_u^{(k)}, y_u^{(k)}) \leq \hat{\delta}, \end{cases}$$

where $\hat{\delta} = \arg\min_{\delta \in \mathbb{R}} \hat{R}(\delta)$. The pseudo-labeled risk is defined as:

$$R_{D_U}(f_\theta) = \frac{1}{n_U} \sum_{k=1}^{n_U} \left[ -\tilde{s}_k \log \sigma \left( r_\theta(x_u^{(k)}, y_u^{(k)}) - \hat{\delta} \right) - (1 - \tilde{s}_k) \log \left( 1 - \sigma \left( r_\theta(x_u^{(k)}, y_u^{(k)}) - \hat{\delta} \right) \right) \right].$$

The gradient with respect to $\theta$ for each $k$-th sample is:

$$\nabla_\theta R_{D_U}(f_\theta)^{(k)} = \begin{cases} -\beta \cdot \underbrace{\mathbb{P}_{D_U}(s=1) \, \sigma(\hat{\delta} - r_\theta(x_u^{(k)}, y_u^{(k)}))}_{\text{Gradient weight}} \cdot \nabla_\theta r_\theta(x_u^{(k)}, y_u^{(k)}) & \text{(if } \tilde{s}_k = 1) \\[2em] -\beta \cdot \underbrace{\mathbb{P}_{D_U}(s=0) \, \sigma(r_\theta(x_u^{(k)}, y_u^{(k)}) - \hat{\delta})}_{\text{Gradient weight}} \cdot \left( -\nabla_\theta r_\theta(x_u^{(k)}, y_u^{(k)}) \right) & \text{(if } \tilde{s}_k = 0). \end{cases}$$

Here, $\beta$ is the same scaling coefficient used in the paired-data risk, and $P_{D_U}(s)$ is the prior for pseudo-labeling. The gradient direction encourages increasing the reward of pseudo-winning samples ($\tilde{s}_k = 1$) and decreasing the reward of pseudo-losing samples ($\tilde{s}_k = 0$), with the magnitude controlled by the sigmoid-based confidence terms $\sigma(\hat{\delta} - r_\theta)$ and $\sigma(r_\theta - \hat{\delta})$. Intuitively, when $r_\theta(x_u^{(k)}, y_u^{(k)})$ is far above the threshold, the model receives a strong signal to further reinforce this response; when it is far below, the model is pushed to suppress it. If pseudo-labels are highly noisy so that $\tilde{s}_k$ is nearly random, the two gradient directions partially cancel in expectation, which limits the impact of erroneous pseudo-labels.

Combining the two components, the full gradient of our objective in Eq. (12) can be written as

$$\nabla_\theta \mathcal{L}(f_\theta) = \gamma' \, \nabla_\theta R_{D_L}(f_\theta) + (1 - \gamma') \, \nabla_\theta R_{D_U}(f_\theta).$$

As discussed in Section 4.3, $\gamma'$ starts from 1 and decays toward $\gamma_{\min}$, so that early updates are dominated by $\nabla_\theta R_{D_L}(f_\theta)$ from human-labeled comparisons, while the influence of $\nabla_\theta R_{D_U}(f_\theta)$ grows only after the reward model has already learned a meaningful separation between winning and losing responses. This interaction between the scheduler and the gradient structure explains why SSPO remains robust even when pseudo-labels are imperfect, as also evidenced by our toy experiments with heavy label noise and our domain-specific experiments in Section 5.2.

## C  IMPLEMENTATION DETAILS

### C.1  IMPLEMENTATION OF PSEUDO-LABELED RISK

While there are many candidates for the reward threshold $\hat{\delta}$ in $R_{D_U}(f_\theta)$, we propose using the Bayes-risk-minimizing threshold for effective pseudo-labeling of unpaired data. To address potential instability in the reward function during training, we normalize the rewards using exponential moving averages (EMA) of the mean and standard deviation.

The pseudo-labeled risk is reformulated as:

$$R_{D_U}(f_\theta) = \frac{1}{n_U} \sum_{k=1}^{n_U} \ell(f_\theta, \tilde{s}_k) \cdot \mathbb{P}_{D_U}(s = \tilde{s}_k) \quad \text{where} \quad \tilde{s}_k = \mathbb{I}\left\{ \frac{r_\theta(x_u^{(k)}, y_u^{(k)}) - \mu_t}{\sigma_t} > \hat{\delta}_t \right\}.$$

The EMA statistics are computed as:

$$\mu_t = m \cdot \mu_{t-1} + (1 - m) \cdot \mu_B, \quad \sigma_t = m \cdot \sigma_{t-1} + (1 - m) \cdot \sigma_B,$$

$$\hat{\delta}_B = \arg\min_{\delta \in \mathbb{R}} \left[ \mathbb{P}_{D_U}(s = 1) \cdot \int_{-\infty}^{\delta} \hat{p}_w^B(r) dr + \mathbb{P}_{D_U}(s = 0) \cdot \int_{\delta}^{\infty} \hat{p}_l^B(r) dr \right],$$

$$\hat{\delta}_t = m \cdot \hat{\delta}_{t-1} + (1 - m) \cdot \hat{\delta}_B.$$

Note that $\hat{p}_w^B$ and $\hat{p}_l^B$ are the estimated density of winning and losing rewards within a minibatch B, respectively. $\mu_B$ and $\sigma_B$ denote the mean and standard deviation of reward values for all responses within the minibatch. We set the momentum parameter $m = 0.95$.

This formulation ensures that the pseudo-labeling threshold remains stable and robust throughout training, even when the reward distribution shifts dynamically. From a theoretical perspective, the KDE-based update of $\hat{\delta}_B$ can be seen as a mini-batch approximation of the global Bayes-risk minimizer in Section 4.2, while the EMA statistics $(\mu_t, \sigma_t, \hat{\delta}_t)$ smooth out stochastic fluctuations across batches. In the limit of many updates, this procedure tracks a slowly varying approximation to the population-level optimal threshold, while remaining numerically stable in the non-stationary regime induced by ongoing policy updates. The combination of theoretically grounded threshold estimation via KDE and practical stabilization through EMA provides both theoretical soundness and training stability.

## C.2 TOY EXPERIMENT

We report full results of test accuracy on the toy dataset without or with noise, with a fixed number of unpaired samples ($n_U$=1,000) in Table 6. SSPO consistently and significantly outperforms all baselines across different quantities of labeled data ($n_L$). The advantage is most pronounced in the extremely data-scarce setting ($n_L$=10), a substantial improvement over the best baselines. This result underscores SSPO's data efficiency, originating in its ability to effectively utilize the large pool of unlabeled data. Furthermore, SSPO's performance is robust, consistently delivering notable results across a broad spectrum of prior probability settings for the unpaired data. Table 7 describes a detailed configuration of toy experiments. The training procedure follows standard practices with 10 epochs, a total batch size of 16, and model selection based on the minimum of validation loss.

## C.3 REAL-DATA EXPERIMENTS

Table 8 describes a detailed configuration of experiments on real-world datasets. Our implementation builds upon the LlamaFactory framework (Zheng et al., 2024), a widely used open-source library for training, fine-tuning, and serving LLMs. We detail the hyperparameter configurations used for each baseline method across the three backbone models. All experiments were conducted using 2 GPUs of A6000, with tuning strategies adjusted per model size. All unspecified hyperparameters for Mistral and Llama3 baselines follow the configurations provided in SimPO[2] (Meng et al., 2024).

We use LoRA (Hu et al., 2021) for rank 8 after exploring ranks from the set {4, 8, 32, 256} for all backbones. We perform a learning rate sweep between 1e-4 and 1e-9 and report results with the best performing value. The total batch size, gradient accumulation steps, and cutoff length are

Table 6: **Comparison of test accuracy on toy dataset across noise levels in paired data.**

| **Baseline** (prior) | $n_L = 10$ | $n_L = 50$ | $n_L = 100$ |
|---|---|---|---|
| *Noise 0%* | | | |
| DPO | 0.743 | 0.777 | 0.846 |
| ORPO | 0.590 | 0.679 | 0.710 |
| SimPO | 0.762 | 0.776 | 0.817 |
| **SSPO** (0.1) | 0.822 | **0.910** | 0.965 |
| **SSPO** (0.3) | 0.810 | 0.889 | 0.955 |
| **SSPO** (0.5) | **0.841** | 0.879 | 0.960 |
| **SSPO** (0.7) | 0.800 | 0.857 | **0.966** |
| **SSPO** (0.9) | 0.840 | 0.861 | 0.965 |
| *Noise 10%* | | | |
| DPO | 0.695 | 0.742 | 0.783 |
| ORPO | 0.595 | 0.646 | 0.682 |
| SimPO | 0.744 | 0.737 | 0.770 |
| **SSPO** (0.5) | **0.840** | **0.812** | **0.931** |
| *Noise 30%* | | | |
| DPO | 0.673 | 0.665 | 0.682 |
| ORPO | 0.601 | 0.617 | 0.627 |
| SimPO | 0.668 | 0.665 | 0.601 |
| **SSPO** (0.5) | **0.698** | **0.739** | **0.733** |
| *Noise 50%* | | | |
| DPO | 0.571 | 0.567 | 0.554 |
| ORPO | 0.594 | 0.586 | 0.549 |
| SimPO | 0.594 | 0.563 | 0.534 |
| **SSPO** (0.5) | **0.757** | **0.656** | **0.563** |

Table 7: **Best hyperparameter settings for the toy experiment.** DPO, ORPO, and SimPO report only the learning rate, while SSPO reports both the learning rate (left) and decay rate (right).

| **Baseline** (prior) | $n_L = 10$ | $n_L = 50$ | $n_L = 100$ |
|---|---|---|---|
| DPO | 1e-3 | 1e-3 | 1e-3 |
| ORPO | 1e-5 | 1e-5 | 1e-5 |
| SimPO | 1e-3 | 1e-3 | 1e-3 |
| **SSPO** (0.1) | 1e-3, 0.01 | 3e-3, 0.001 | 1e-3, 0.03 |
| **SSPO** (0.3) | 5e-3, 0.001 | 1e-3, 0.03 | 5e-3, 0.05 |
| **SSPO** (0.5) | 2e-2, 0.03 | 1e-3, 0.001 | 5e-3, 0.005 |
| **SSPO** (0.7) | 1e-2, 0.01 | 1e-3, 0.001 | 1e-3, 0.001 |
| **SSPO** (0.9) | 5e-3, 0.001 | 1e-3, 0.03 | 5e-4, 0.001 |

[2]https://github.com/princeton-nlp/SimPO

Table 8: **Hyperparameter settings for real-data experiments across all domains and backbones.** All the algorithms are configured with prior 0.5 under two different data sizes.

| Method | Size | Hyperparameter | UltraFeedback | | | UltraMedical-Preference | | DSP Business | |
|---|---|---|---|---|---|---|---|---|---|
| | | | Phi-2 | Mistral | Llama3 | Mistral | Llama3 | Mistral | Llama3 |
| DPO | 1% | Training Epochs | 1 | 1 | 1 | 1 | 1 | 1 | 1 |
| | | Learning Rate | 1e-6 | 5e-7 | 1e-6 | 5e-7 | 5e-7 | 5e-7 | 1e-6 |
| | | $\beta$ | 0.1 | 0.1 | 0.1 | 0.1 | 0.1 | 0.1 | 0.1 |
| | 10% | Training Epochs | 1 | 1 | 1 | 1 | 1 | 1 | 1 |
| | | Learning Rate | 1e-6 | 5e-7 | 1e-6 | 5e-7 | 5e-7 | 5e-6 | 1e-6 |
| | | $\beta$ | 0.1 | 0.1 | 0.1 | 0.1 | 0.1 | 0.1 | 0.1 |
| ORPO | 1% | Training Epochs | 1 | 1 | 1 | 1 | 1 | 1 | 1 |
| | | Learning Rate | 5e-6 | 5e-6 | 5e-6 | 3e-7 | 3e-7 | 5e-6 | 5e-6 |
| | | $\lambda$ | 0.25 | 0.2 | 0.2 | 0.2 | 0.2 | 0.2 | 0.2 |
| | 10% | Training Epochs | 1 | 1 | 1 | 1 | 1 | 1 | 1 |
| | | Learning Rate | 7e-6 | 5e-6 | 5e-6 | 3e-7 | 3e-7 | 1e-6 | 5e-6 |
| | | $\lambda$ | 0.25 | 0.2 | 0.2 | 0.2 | 0.2 | 0.25 | 0.2 |
| SimPO | 1% | Training Epochs | 1 | 1 | 1 | 1 | 1 | 1 | 1 |
| | | Learning Rate | 5e-7 | 5e-7 | 6e-7 | 5e-7 | 5e-7 | 6e-7 | 6e-7 |
| | | $\beta$ | 10 | 2 | 2 | 2 | 2 | 2 | 2 |
| | | $\Delta$ | 2 | 0.8 | 1 | 0.8 | 0.8 | 1 | 1 |
| | 10% | Training Epochs | 1 | 1 | 1 | 1 | 1 | 1 | 1 |
| | | Learning Rate | 5e-7 | 5e-7 | 1.5e-7 | 5e-7 | 5e-7 | 5e-7 | 6e-7 |
| | | $\beta$ | 10 | 2 | 2 | 2 | 2 | 2 | 2 |
| | | $\Delta$ | 2 | 0.8 | 1 | 0.8 | 0.8 | 0.8 | 1 |
| SSRM | 1% | Training Epochs | 1 | 1 | 1 | 1 | 1 | 1 | 1 |
| | | Learning Rate | 5e-7 | 5e-7 | 6e-7 | 5e-7 | 5e-7 | 6e-7 | 6e-7 |
| | | $\beta$ | 10 | 2 | 2 | 2 | 2 | 2 | 2 |
| | | $\Delta$ | 2 | 0.8 | 1 | 0.8 | 0.8 | 1 | 1 |
| | 10% | Training Epochs | 1 | 1 | 1 | 1 | 1 | 1 | 1 |
| | | Learning Rate | 5e-7 | 5e-7 | 6e-7 | 5e-7 | 5e-7 | 6e-7 | 6e-7 |
| | | $\beta$ | 10 | 2 | 2 | 2 | 2 | 2 | 2 |
| | | $\Delta$ | 2 | 0.8 | 1 | 0.8 | 0.8 | 1 | 1 |
| KTO | 1% | Training Epochs | 1 | 1 | 1 | 1 | 1 | 1 | 1 |
| | | Learning Rate | 5e-6 | 5e-6 | 5e-6 | 1e-6 | 1e-6 | 1e-6 | 1e-6 |
| | 10% | Training Epochs | 1 | 1 | 1 | 1 | 1 | 1 | 1 |
| | | Learning Rate | 5e-6 | 5e-6 | 5e-6 | 1e-6 | 1e-6 | 1e-6 | 1e-6 |
| SPA | 1% | Training Epochs | 1 | 1 | 1 | 1 | 1 | 1 | 1 |
| | | Learning Rate | 5e-6 | 1e-5 | 1e-5 | 1e-5 | 1e-5 | 1e-5 | 1e-5 |
| | | $\beta$ | 0.01 | 0.01 | 0.01 | 0.01 | 0.01 | 0.01 | 0.01 |
| | | Iteration | 1 | 2 | 2 | 2 | 2 | 2 | 2 |
| | 10% | Training Epochs | 1 | 1 | 1 | 1 | 1 | 1 | 1 |
| | | Learning Rate | 5e-6 | 2e-5 | 1e-5 | 1e-5 | 1e-5 | 2e-5 | 1e-5 |
| | | $\beta$ | 0.01 | 0.01 | 0.01 | 0.01 | 0.01 | 0.01 | 0.01 |
| | | Iteration | 1 | 2 | 2 | 2 | 2 | 2 | 2 |
| **SSPO** | 1% | Training Epochs | 2 | 2 | 1 | 1 | 1 | 3 | 3 |
| | | Learning Rate | 1e-4 | 1e-5 | 1e-5 | 5e-5 | 1e-5 | 5e-5 | 1e-5 |
| | | Decay Rate ($\lambda$) | 0.005 | 0.001 | 0.005 | 0.01 | 0.001 | 0.001 | 0.001 |
| | 10% | Training Epochs | 1 | 1 | 2 | 1 | 1 | 1 | 1 |
| | | Learning Rate | 1e-6 | 1e-5 | 1e-5 | 5e-6 | 5e-6 | 5e-5 | 5e-5 |
| | | Decay Rate ($\lambda$) | 0.01 | 0.001 | 0.01 | 0.001 | 0.01 | 0.001 | 0.001 |

---

**Algorithm 1** Semi-Supervised Preference Optimization (**SSPO**)

---

**Require:** Paired dataset $D_L = \{(x^{(i)}, y_w^{(i)}, y_l^{(i)})\}$, Unpaired dataset $D_U = \{(x_u^{(j)}, y_u^{(j)})\}$
**Require:** Scheduler parameters $\gamma_0, \gamma_{\min}$, decay rate $\lambda > 0$
**Require:** Momentum $m \in (0, 1)$, total training steps $\mathcal{T}$

 1: Initialize EMA stats: $\mu_0, \sigma_0, \hat{\delta}_0$

 2: **for** $t = 1$ to $\mathcal{T}$ **do**
 3:      Sample minibatch $B_L \subset D_L$, $B_U \subset D_U$
 4:      Compute rewards $r_\theta(x, y)$ for all $(x, y) \in B_L \cup B_U$:
 5:          $\mu_B \leftarrow \text{mean}(\{r_\theta(x, y) \mid (x, y) \in B_L \cup B_U\})$
 6:          $\sigma_B \leftarrow \text{std}(\{r_\theta(x, y) \mid (x, y) \in B_L \cup B_U\})$
 7:          $\mu_t \leftarrow m \cdot \mu_{t-1} + (1 - m) \cdot \mu_B$
 8:          $\sigma_t \leftarrow m \cdot \sigma_{t-1} + (1 - m) \cdot \sigma_B$
 9:      Estimate $\hat{p}_w^B(r)$ and $\hat{p}_l^B(r)$ using KDE on $B_L$:
10:          $\hat{\delta}_B \leftarrow \arg\min_\delta \left[ \mathbb{P}_{D_U}(s = 1) \int_{-\infty}^{\delta} \hat{p}_w^B(r) dr + \mathbb{P}_{D_U}(s = 0) \int_{\delta}^{\infty} \hat{p}_l^B(r) dr \right]$
11:          $\hat{\delta}_t \leftarrow m \cdot \hat{\delta}_{t-1} + (1 - m) \cdot \hat{\delta}_B$
12:      **for all** $(x_u, y_u) \in B_U$ **do**
13:          $r_{\text{norm}}(x_u, y_u) \leftarrow \frac{r_\theta(x_u, y_u) - \mu_t}{\sigma_t}$
14:          **if** $r_{\text{norm}}(x_u, y_u) > \hat{\delta}_t$ **then**
15:            Assign pseudo-label $\tilde{s} = 1$
16:          **else**
17:            Assign pseudo-label $\tilde{s} = 0$
18:          **end if**
19:      **end for**
20:      $\gamma_t \leftarrow \max(\gamma_{\min}, \gamma_0 \cdot \exp(-\lambda t))$
21:      $\mathcal{L}(f_\theta) \leftarrow \gamma_t \cdot R_{B_L}(f_\theta) + (1 - \gamma_t) \cdot R_{B_U}(f_\theta)$
22:      Update $f_\theta$ using gradient descent
23: **end for**

---

fixed at 64, 8, and 1024, respectively, across all models. We select the best model based on the minimum validation loss.

For SSRM, we follow the original protocol by using a preset confidence threshold and generating response pairs from prompts with no annotation for pseudo-labeling after training the reward model[3]. To ensure fair comparison, we generate one additional response using the SFT model of each backbone and use the reward model at each data size. We run three iterations for each setting with a default confidence level of 0.8, meaning the model incorporates new data pairs only when they exceed this confidence threshold. For SSPO, we fix $\beta = 10$ and $\Delta = 2$ and the adaptive scheduler $\gamma'$ is bounded below by $\gamma_{\min} = n_L / (n_L + n_U)$. We set the pseudo-labeling prior to 0.5 since there is no suitable preference information available for the UltraChat.

### C.4 PSEUDO-CODE OF SSPO

Algorithm 1 presents the pseudo-code for training SSPO with both paired and unpaired data. The algorithm begins by computing the SimPO loss on preference-labeled pairs, followed by generating pseudo-labels for unpaired responses based on a reward threshold. The threshold is determined using the Bayes-risk-minimizing approach via KDE estimation on the current minibatch. To improve stability and consistency across training steps, we standardize the rewards and apply EMA to the statistics and threshold values. This practical design helps smooth out fluctuations in threshold computation and ensures reliable pseudo-labeling decisions. The final training loss is computed as a weighted combination of the paired data and pseudo-labeled risks, where the weight is controlled by a scheduling function that gradually increases the contribution of unpaired data.

---

[3]https://github.com/RLHFlow/RLHF-Reward-Modeling

# D  THE DIFFERENCE BETWEEN SSRM, SPA, AND SSPO

In this section, we provide a detailed comparison between SSPO and two prominent semi-supervised alignment frameworks: Semi-Supervised Reward Modeling (**SSRM**) (He et al., 2024) and Spread Preference Annotation (**SPA**) (Kim et al., 2025). While all three methods aim to leverage unlabeled data to mitigate the scarcity of human preference labels, SSPO distinguishes itself through its theoretical grounding, data efficiency, and non-iterative training paradigm. We analyze these methods across four key dimensions: Data Composition, Algorithmic Formulation, Theoretical Foundation, and Computational Efficiency.

## D.1  DATA COMPOSITION & ANNOTATION

Let $x$ be a prompt (i.e., an input of our model $\pi_\theta$ to be trained) and $y$ be a corresponding response (i.e., an output of our model $\pi_\theta$). All three algorithms leverage both labeled and unlabeled datasets, but the way to formulate and generate the datasets is quite different.

- **SSRM:** Requires a small set of paired and labeled data $D_L = \{(x^{(i)}, y_w^{(i)}, y_l^{(i)})\}_{i=1}^{n_L}$ and a large set of *unlabeled pairs* $D_U = \{(x_u^{(i)}, y_{u1}^{(i)}, y_{u2}^{(i)})\}_{i=1}^{n_U}$. It relies on the model's confidence to pseudo-label these pairs, essentially performing self-training on an existing paired structure.

- **SPA:** Utilizes a small seed of paired data $D_L = \{(x^{(i)}, y_w^{(i)}, y_l^{(i)})\}_{i=1}^{n_L}$ and assumes that there exists a set of unseen prompts $X_t = \{x_u^{(t,i)}\}_{i=1}^{n_U^{(t)}}$ per $t$-th iteration. It explicitly *generates* multiple responses for the unseen prompts during training to form *new unlabeled pairs* $D_U^{(t)} = \{(x_u^{(t,i)}, y_{u1}^{(t,i)}, y_{u2}^{(t,i)})\}_{i=1}^{n_U^{(t)}}$ per $t$-th iteration, which are then self-labeled using the model's implicit reward by DPO (Rafailov et al., 2024).

- **SSPO (Ours):** Requires a small set of paired data $D_L = \{(x^{(i)}, y_w^{(i)}, y_l^{(i)})\}_{i=1}^{n_L}$ and a large pool of *unpaired and unlabeled data* $D_U = \{(x_u^{(i)}, y_u^{(i)})\}_{i=1}^{n_U}$. Unlike SSRM, it does not require pre-existing pairs in $D_U$, and unlike SPA, it does not require generating new responses during training. SSPO uniquely assigns "winning" or "losing" pseudo-labels to single responses based on a learned reward threshold $\delta$, allowing for the direct utilization of standard SFT datasets.

## D.2  ALGORITHM & FORMULATION

- **SSRM & SPA:** Both methods employ an iterative cycle: Training $\rightarrow$ Inference/Generation $\rightarrow$ Pseudo-labeling $\rightarrow$ Filtering $\rightarrow$ Re-training. SSRM treats reward modeling as a classification task, while SPA uses the DPO objective for self-refinement. This iterative process is inherently complex and sensitive to the quality of intermediate models.

- **SSPO (Ours):** SSPO frames preference learning as a Bayes-optimal classification problem. It optimizes a joint objective combining a standard preference loss on $D_L$ and a threshold-based binary classification loss on $D_U$: $\mathcal{L}(f_\theta) = \gamma' \cdot R_{D_L}(f_\theta) + (1 - \gamma') \cdot R_{D_U}(f_\theta)$ (Eq. (12)). By dynamically estimating the optimal threshold $\hat{\delta}$ via Kernel Density Estimation (KDE) and using an adaptive scheduler $\gamma'$, SSPO achieves robust alignment in a single training stage without iterative loops.

## D.3  THEORETICAL FOUNDATION

- **SSRM & SPA:** Primarily rely on empirical validation of self-training principles (e.g., noisy label learning, curriculum learning) without specific theoretical guarantees for the optimality of their pseudo-labeling strategies in the context of preference distributions.

- **SSPO (Ours):** Provides a rigorous theoretical foundation (Theorem 1). We prove the existence of an optimal reward threshold $\delta^*$ that separates winning and losing response distributions with high probability under sub-Gaussian assumptions. This theoretical guarantee underpins our pseudo-labeling strategy, ensuring it is statistically principled rather than heuristic.

### D.4 COMPUTATIONAL EFFICIENCY

- **SSRM & SPA:** Computationally expensive due to their iterative nature. They require multiple rounds of inference over the entire unlabeled dataset (and generation for SPA), followed by re-training.

- **SSPO (Ours):** Highly efficient single-stage training. The additional cost of KDE and threshold estimation is negligible compared to the overhead of iterative generation and inference. SSPO reuses existing SFT datasets directly, eliminating the need for costly response generation during the alignment phase. We provide more analyses of time complexity in Appendix E.

## E COMPUTATIONAL OVERHEAD

### E.1 TIME COMPLEXITY OF SSPO

In this section, we give a simple big-O analysis of the computational overhead introduced by SSPO. Let $D_L$ denote the paired dataset and $D_U$ the unpaired dataset. At each training step $t$, Algorithm 1 samples a minibatch $B_L \subset D_L$ and $B_U \subset D_U$ and computes rewards for all responses in $B_L \cup B_U$. Let $N_{\text{tok}}$ be the total number of tokens in the current batch (including instructions and all responses), and let $d$ be the model dimension. The dominant cost per step comes from the forward and backward passes of the transformer model, which scale as $\mathcal{O}(N_{\text{tok}} \cdot d^2)$ for a single causal decoder, or $\mathcal{O}(N_{\text{tok}} \cdot H \cdot d^2)$ if we make the number of attention heads $H$ explicit. When a separate reference model is used, this cost is increased by a constant factor (roughly $\times 2$), but the asymptotic order remains unchanged.

SSPO adds a few extra components on top of this base cost:

- **Reward standardization and EMA updates.** For a batch of $|B_L| + |B_U|$ rewards, computing the batch mean and variance, and updating the exponential moving averages $\mu_t$ and $\sigma_t$ costs $\mathcal{O}(|B_L| + |B_U|)$.

- **KDE-based Bayes-risk threshold.** Let $M = |B_L|$ be the number of labeled pairs in the batch, and let $G$ be the number of grid points used to evaluate the kernel density estimators (we use $G = 200$ in practice). The complexity of fitting Gaussian KDEs for the winning and losing rewards and searching for the Bayes-risk-minimizing threshold is $\mathcal{O}(M^2 + MG)$, which we can conservatively write as $\mathcal{O}(M^2)$ with respect to the batch size.

- **Pseudo-labeling of unpaired responses.** For each unpaired response $(x_u, y_u) \in B_U$, SSPO compares the normalized reward to the current threshold and computes a logistic-style loss. This step scales linearly in the number of unpaired items, i.e., $\mathcal{O}(|B_U|)$.

Putting these terms together, the per-step complexity of SSPO can be summarized as

$$\mathcal{O}(N_{\text{tok}} \cdot d^2) \; + \; \mathcal{O}(|B_L| + |B_U|) \; + \; \mathcal{O}(M^2) \; \approx \; \mathcal{O}(N_{\text{tok}} \cdot d^2),$$

where the transformer forward/backward passes clearly dominate. Over $\mathcal{T}$ training steps with a total of $N_{\text{tok}}$ processed tokens, the overall complexity becomes

$$\mathcal{O}(N_{\text{tok}} \cdot d^2).$$

In other words, SSPO has the same asymptotic time complexity as DPO and SimPO; the KDE-based thresholding and pseudo-labeling introduce only a modest constant-factor overhead that is negligible compared to the cost of running LLMs. Likewise, DPO incurs a slightly larger constant factor due to the additional reference-model forward pass, and both SimPO and SSPO yield the smallest constant factor because they require only a single policy-model forward/backward pass without any reference model.

### E.2 EMPIRICAL TRAINING TIME OF SSPO

We also report the empirical training time required to run SSPO under our experimental setup. Table 9 summarizes the observed training time *per epoch* and the average number of samples processed per second. All backbones are trained with LoRA adapters with rank 8 on 2 GPUs of A6000 with a total batch size of 64. Combined with the complexity analysis in Appendix E.1, these measurements indicate that SSPO is practically feasible.

Table 9: **Per-epoch training runtime (Runtime) and training samples per second (Samples/s) of SSPO across backbones and paired-data sizes.** UltraChat is always used at 10% ($n_U$=20,786) of its dataset, while UltraFeedback is used at either 1% ($n_L$=611) or 10% ($n_L$=6,113).

| Size | Phi-2 (2.7B) | | Mistral (7B) | | Llama3 (8B) | |
|------|---------|----------|---------|----------|---------|----------|
| | Runtime | Samples/s | Runtime | Samples/s | Runtime | Samples/s |
| 1% | 4,764s **(1.3h)** | 4.47 | 9,684s **(2.7h)** | 2.20 | 9,282s **(2.6h)** | 2.29 |
| 10% | 7,813s **(2.2h)** | 3.36 | 19,350s **(5.4h)** | 1.35 | 15,919s **(4.4h)** | 1.65 |

## F  MORE QUANTITATIVE ANALYSES

### F.1  DPO AND SIMPO COMBINED WITH UNPAIRED DATA

For further comparison, we extend the preference optimization framework to two practical variants that jointly leverage paired and unpaired data. We combine a standard preference objective on paired data with a supervised fine-tuning (SFT) objective on unpaired data to both DPO and SimPO objectives to compare the performance with the same setting of data utilization. We call these variants **DPO+SFT** and **SimPO+SFT**. Both methods follow the same data formulation as SSPO but differ in the formulation of loss.

**Setup.** We denote by $\pi_\theta(y \mid x)$ the policy model and by $\pi_{\text{ref}}(y \mid x)$ a fixed reference model when required. Given a pair $(x, y_w, y_l)$ of the paired dataset $D_L$, the DPO objective can be written as

$$\mathcal{L}_{\text{DPO}}(\theta; x, y_w, y_l) = -\log \sigma \Big[ \beta \big( \log \frac{\pi_\theta(y_w \mid x)}{\pi_{\text{ref}}(y_w \mid x)} - \log \frac{\pi_\theta(y_l \mid x)}{\pi_{\text{ref}}(y_l \mid x)} \big) \Big],$$

and the corresponding expected risk over $D_L$ is $R_{D_L}^{\text{DPO}}(\theta) = \mathbb{E}_{(x,y_w,y_l) \sim D_L} \big[ \mathcal{L}_{\text{DPO}}(\theta; x, y_w, y_l) \big]$.

To incorporate abundant unpaired data $D_U$, we define a supervised fine-tuning loss that treats each $(x_u, y_u)$ as winning sample:

$$\mathcal{L}_{\text{SFT}}(\theta; x_u, y_u) = -\log \pi_\theta(y_u \mid x_u),$$

and the corresponding unpaired risk is $R_{D_U}^{\text{SFT}}(\theta) = \mathbb{E}_{(x_u,y_u) \sim D_U} \big[ \mathcal{L}_{\text{SFT}}(\theta; x_u, y_u) \big]$. The DPO+SFT objective is then defined as a weighted sum of the preference risk on $D_L$ and the SFT risk on $D_U$:

$$\mathcal{L}_{\text{DPO+SFT}}(\theta) = R_{D_L}^{\text{DPO}}(\theta) + \lambda_{\text{SFT}} \cdot R_{D_U}^{\text{SFT}}(\theta),$$

where $\lambda_{\text{SFT}} \geq 0$ controls the relative importance of the unpaired dataset term. When $\lambda_{\text{SFT}} = 0$, the objective reduces to standard DPO.

Similarly, using the reward function with length normalization and a prescribed margin $\Delta$, the SimPO loss on a single pair $(x, y_w, y_l)$ is

$$\mathcal{L}_{\text{SimPO}}(\theta; x, y_w, y_l) = -\log \sigma \left[ \frac{\beta}{|y_w|} \log \pi_\theta(y_w \mid x) - \frac{\beta}{|y_l|} \log \pi_\theta(y_l \mid x) - \Delta \right],$$

and the corresponding risk over $D_L$ is $R_{D_L}^{\text{SimPO}}(\theta) = \mathbb{E}_{(x,y_w,y_l) \sim D_L} \big[ \mathcal{L}_{\text{SimPO}}(\theta; x, y_w, y_l) \big]$.

Analogously to DPO+SFT, we reuse the SFT loss on $D_U$ added with the length normalization:

$$R_{D_U}^{\text{SFT}'}(\theta) = \mathbb{E}_{(x_u,y_u) \sim D_U} \left[ -\frac{1}{|y_u|} \log \pi_\theta(y_u \mid x_u) \right].$$

Therefore, the SimPO+SFT objective is then given by

$$\mathcal{L}_{\text{SimPO+SFT}}(\theta) = R_{D_L}^{\text{SimPO}}(\theta) + \lambda_{\text{SFT}} \cdot R_{D_U}^{\text{SFT}'}(\theta),$$

using the same SFT coefficient $\lambda_{\text{SFT}}$. This formulation yields a reference-free variant that still benefits from unpaired data.

**Results.** Table 10 presents the comparative results on the UltraFeedback dataset. While the SFT-augmented baselines show improvements over their standard counterparts by leveraging the additional unpaired data, SSPO consistently outperforms all variants across different data sizes. This performance gap highlights the critical advantage of our algorithmic design. DPO+SFT and SimPO+SFT treat the unpaired data $D_U$ merely as a collection of positive demonstrations, applying a uniform SFT objective to all samples. This approach essentially regularizes the policy towards the distribution of the unpaired corpus but fails to distinguish the varying quality or preference alignment within that data. In contrast, SSPO does not indiscriminately reinforce all unpaired responses. By employing a dynamically learned reward threshold, SSPO actively performs *pseudo-labeling*, effectively filtering and differentiating pseudo-winning responses from pseudo-losing ones. This mechanism allows the model to learn a more nuanced preference boundary rather than simply imitating the aggregate statistics of the unpaired dataset. Therefore, the superior performance of SSPO is not merely a result of data scale, but a direct consequence of its ability to extract and leverage latent preference signals through principled semi-supervised learning.

Table 10: **Comparing the performance of DPO+SFT and SimPO+SFT with SSPO at UltraFeedback.** We set $\lambda_{\text{SFT}} = 1$ for both DPO+SFT and SimPO+SFT. The best numbers are in bold, and the second-best ones are underlined.

| Baseline | Size | Mistral (7B) | | |
|---|---|---|---|---|
| | | LC | WR | MT |
| DPO+SFT | 1% | 26.5 | 17.0 | 7.6 |
| | 10% | 29.5 | 19.5 | **7.7** |
| SimPO+SFT | 1% | 25.1 | 16.6 | 7.6 |
| | 10% | 27.3 | 17.3 | 7.6 |
| **SSPO** | 1% | **26.7** | **18.1** | 7.7 |
| | 10% | **30.0** | **20.7** | 7.7 |

## F.2 PERFORMANCES WITH MUCH PAIRED DATA

We further investigated the performance of SSPO in regimes where paired preference data is abundant, to determine if the benefits of semi-supervised learning persist when the scarcity constraint is lifted. We conducted experiments on both the synthetic toy dataset and the real-world UltraFeedback dataset by scaling up the amount of paired data.

Table 11 shows the test accuracy on the toy dataset as the number of labeled pairs ($n_L$) increases to 1.000 and 10,000. As expected, all methods converge towards high accuracy as the volume of supervision increases. This confirms that the $R_{D_U}(f_\theta)$ does not conflict with $R_{D_L}(f_\theta)$ but rather complements it, ensuring stability and high performance regardless of the size of paired data.

Table 11: **Comparison of test accuracy on toy dataset without noise in different paired data.** The best numbers for each size of $n_L$ are in bold, and the second-best ones are underlined.

| Method | $n_L = 10$ | $n_L = 50$ | $n_L = 100$ | $n_L = 1,000$ | $n_L = 10,000$ |
|---|---|---|---|---|---|
| DPO | 0.743 | 0.777 | 0.846 | **0.986** | 0.995 |
| ORPO | 0.590 | 0.679 | 0.710 | 0.832 | 0.934 |
| SimPO | 0.762 | 0.776 | 0.817 | 0.981 | **0.996** |
| **SSPO** (0.1) | 0.822 | **0.910** | 0.965 | **0.986** | 0.992 |
| **SSPO** (0.3) | 0.810 | 0.889 | 0.955 | 0.983 | 0.993 |
| **SSPO** (0.5) | **0.841** | 0.879 | 0.960 | 0.983 | **0.996** |
| **SSPO** (0.7) | 0.800 | 0.857 | **0.966** | 0.985 | 0.991 |
| **SSPO** (0.9) | 0.840 | 0.861 | 0.965 | **0.986** | 0.991 |

In addition, Table 12 presents the results on the full UltraFeedback dataset (100% of data, $n_L$=61,135). Notably, SSPO trained on the complete labeled dataset achieves LC of **32.4%**, significantly outperforming the strong semi-supervised baseline SPA (26.2%) trained on the same amount of labeled data. Furthermore, this performance is superior to SSPO trained on 10% data (30.0%), indicating that the model continues to benefit from the additional labeled data. This result suggests that even when ground-truth labels are sufficient, the principled pseudo-labeling of unpaired data provides signals that further refine the policy beyond what is possible with supervised signals alone.

Table 12: **Comparing the performance when training with 100% of UltraFeedback.** The best numbers are in bold, and the second-best ones are underlined.

| Baseline | Size | Mistral (7B) | | |
|---|---|---|---|---|
| | | LC | WR | MT |
| SPA | 1% | 18.2 | 15.6 | **7.7** |
| | 10% | 19.1 | 18.7 | **7.8** |
| | 100% | 26.2 | 19.3 | 7.7 |
| **SSPO** | 1% | **26.7** | **18.1** | **7.7** |
| | 10% | **30.0** | **20.7** | 7.7 |
| | 100% | **32.4** | **21.0** | **7.8** |

# G   QUALITATIVE EXAMPLES

## G.1   MORE CASE STUDIES

In Section 5.4, we conducted an analysis using a substring matching algorithm to find semantically similar content between evaluation questions and our unpaired datasets to identify the specific benefits of unpaired data. Table 13 shows how SSPO learns from the well-structured format of an unpaired medical guide to produce a multifaceted response about stress management, whereas the baseline provides a single block of text. Similarly, Table 14 demonstrates SSPO's ability to internalize a professional tone and actionable list format from the business-domain data, successfully fulfilling a complex persona-based request that the baseline fails to comprehend. In both cases, SSPO successfully aligns its generation based on high-quality stylistic and structural cues from unpaired responses, leading to superior outputs.

## G.2   QUALITATIVE COMPARISON

Tables 15–19 present qualitative comparisons across both general and specialized prompts, including open-ended questions and domain-specific explanations. These examples illustrate the differences in content structure, informativeness, and stylistic alignment between SSPO and other preference optimization methods.

**Context-aware reasoning with balanced informativeness.**   In open-ended questions such as "What is the airspeed velocity of an unladen swallow?" (Table 15), SSPO stands out by maintaining a clear and concise structure while contextualizing the question accurately. It references the film "Monty Python and the Holy Grail," distinguishes humor from fact, and provides an empirical estimate, resulting in an answer that is both informative and appropriately styled. In contrast, DPO and ORPO tend to over-elaborate with tangential details, while SimPO oversimplifies by dismissing the question altogether. In domain-specific prompts (from Table 16 to 19), SSPO continues to provide well-structured responses that surpass the superficial summaries of baselines, whether by presenting multi-step biological mechanisms with accurate terminology or by delivering comprehensive, actionable guides for brand marketing.

**Structural coherence and domain adaptability.**   Across diverse inputs, SSPO demonstrates a consistent ability to organize responses around key conceptual stages, especially in specialized domains such as medicine. Where other methods often plateau at surface-level summaries or fall into redundancy (as with SSRM), SSPO maintains logical progression and avoids repetition. Its answers are technical enough for experts by explicitly structuring the mechanisms (e.g., antigen presentation, immune activation) and aligning stylistic tone with the question's context. These results suggest that SSPO's use of unpaired data not only improves human alignment but also enhances the structural and semantic quality of domain-specific responses.

Table 13: **A case study showing how SSPO benefits from semantically relevant but unpaired data** (Mistral trained on 10% of **UltraMedical-Preference**). The selected response exhibits implicit preference (via a pseudo-label reward), enabling SSPO to generalize effectively in specialized domains like medicine. By learning from the comprehensive, categorized structure of the unpaired medical data, SSPO generates a response that is not just a simple list but a well-organized, multi-faceted guide to stress management, demonstrating effective knowledge transfer. For brevity, some contents are omitted.

| Item | Content |
|---|---|
| **Question** | What are the most effective ways to deal with stress? |
| **KTO** | Stress, defined as a physiological response to a perceived threat, is primarily mediated by the hypothalamic-pituitary-adrenal (HPA) axis, leading to an increase in glucocorticoids such as cortisol. *(omitted)* It is recommended to consult a healthcare professional for a full diagnostic workup and a tailored treatment plan based on your specific stress-related symptomatology. |
| **Pseudo-Winning Datum** (UltraMedical) | **Instruction:** Offer detailed guidance for managing stress and promoting better mental health. 
 **Response:** Managing stress and promoting better mental health are critical aspects of living a balanced and fulfilling life. Here are several detailed strategies to help you navigate through stressful periods and maintain your mental well-being: ### 1. **Practice Mindfulness and Meditation** Mindfulness involves staying present and fully engaging with the moment without judgment. Meditation can help calm your mind, reduce stress, and improve concentration. *(omitted)* ### 2. **Regular Physical Activity** Exercise is a powerful stress reliever. It can improve your mood, enhance your physical health, and increase your energy levels. *(omitted)* |
| **SSPO** (Ours) | 1. **Adrenaline Outlets**: Engaging in activities that involve physical exertion and the release of adrenaline provides an immediate outlet for stress. This can include sports, exercise routines that push personal boundaries, and adventurous pursuits. *(omitted)* 2. **Cognitive Control**: Psychological techniques aimed at managing the perception and interpretation of stress have proven efficacy. *(omitted)* Therefore, an integrative approach involving multiple strategies and healthcare professionals is often the most effective course of treatment and prevention for individuals experiencing severe or chronic stress problems. |

Table 14: **A case study showing how SSPO benefits from semantically relevant but unpaired data** (Mistral trained on 10% of **DSP Business**). While the baseline (KTO) produces an unstructured, keyword-based output that fails to adopt the requested persona, SSPO effectively learns the professional tone and structured format prevalent in the unpaired business dataset. As a result, SSPO generates a clear, actionable, and well-organized response, directly providing the mentee with a comprehensive, enumerated guide. For brevity, some contents are omitted.

| Item | Content |
|---|---|
| **Question** | Pretend to be a business professional. Act like you are giving your mentee professional advice on the most essential points to create a self-sustaining business within the next five years time without the help of external investors. |
| **KTO** | ```[SYSTEM_QUERY]: "self-sustaining business" + "no investors"```
```[RESPONSE_PROTOCOL]: DATA_DUMP```
```INITIATING...```
```KEY_TERMS_LIST:```
```ROI (Return on Investment)```
```GAAP (Generally Accepted Accounting Principles)```
*(omitted)* |
| **Pseudo-Winning Datum** (17k Business Book) | **Instruction:** how can recognizing customer passions ignite word-of-mouth movements? 
 **Response:** Truly understanding customer passions, like by listening to stories about their lives, allows a business to inspire behaviors and tap into causes that customers want to spread. This recognition of passion can ignite self-sustaining word-of-mouth movements. |
| **SSPO** (Ours) | As a business professional, I would advise you to follow these essential points to create a self-sustaining business within the next five years: 1. Develop a comprehensive business plan: Your business plan should outline your mission, vision, goals, target audience, market analysis, marketing strategy, and financial projections.*(omitted)* 2. Conduct thorough market research: Before starting your business, it is important to conduct thorough market research to understand your target audience, competition, and industry trends. *(omitted)* |

Table 15: **Comparison of model responses to the general-domain question from AlpacaEval2.0.** All models are based on **Mistral** and trained with **10% of UltraFeedback**. The table highlights differences in context awareness, informativeness, and stylistic alignment across preference optimization methods. SSPO uniquely balances humor and factual accuracy, while baselines either over-explain or oversimplify.

| Instruction | What is the airspeed velocity of an unladen swallow? |
| --- | --- |
| **DPO** | The question is a famous line from the Monty Python film "Monty Python and the Holy Grail." In the film, the answer is given as 11 meters per second, or about 25 miles per hour. However, this answer is a joke, as the film never actually provides an accurate or scientifically sound answer to this question. In reality, the airspeed velocity of an unladen swallow depends on the species of the swallow, and can range from about 5 to 11 meters per second. For instance, the European Swallow (Hirundo rustica) can reach speeds of up to 11 m/s, while the African Swallow (Hirundo lucida) is typically slower, at around 7 m/s. So, while the Monty Python answer is amusing, it is not a scientifically accurate answer to the question. |
| **ORPO** | The question is a reference to a scene in the movie "Monty Python and the Holy Grail" where the answer is humorously given as "11 meters per second, or 40 kilometers per hour, or 14 miles per hour, or 11 meters per second." The actual answer is not provided in the movie and varies depending on the source, with estimates ranging from 1.5 to 11 meters per second. In reality, the airspeed velocity of a European Swallow is around 11 meters per second (40 kilometers per hour or 14 miles per hour) during level flight, but this assumes the swallow is carrying food. When unladen, the European Swallow can fly at speeds of around 7.5 to 11 meters per second (27 to 40 kilometers per hour or 17 to 25 miles per hour). The African Swallow, on the other hand, has a higher cruising speed of around 12.7 meters per second (45 kilometers per hour or 28 miles per hour). |
| **SimPO** | The question is a popular reference to the film "Monty Python and the Holy Grail." However, the film never actually gives a definitive answer to this question. It is generally accepted that the question itself is a nonsensical one, as swallows do not fly with no weight in their bellies. Therefore, there is no scientifically accurate answer to this question. |
| **SSRM** | 650.
I don't remember learning the airspeed velocity of an unladen swallow in school. But it does make for a great punchline. |
| **KTO** | That's the classic Monty Python question. Everyone knows the correct answer is 100 miles per hour, because otherwise the bridge keeper wouldn't have asked it. So clearly, an unladen swallow must fly at that speed. |
| **SPA** | This question is a well-known line from the movie "Monty Python and the Holy Grail." In the film, a humorous answer of 11 meters per second, or roughly 25 miles per hour, is mentioned. However, this is meant as a joke. The movie does not actually offer a scientifically valid response. In fact, the true airspeed of an unladen swallow varies depending on the species. For example, the European Swallow (Hirundo rustica) can fly at speeds up to 11 m/s, while the African Swallow (Hirundo lucida) tends to fly slower, around 7 m/s. So although the Monty Python reference is entertaining, it shouldn't be taken as a factual answer. |
| **SSPO** | The question "What is the airspeed velocity of an unladen swallow?" is a popular one due to its mention in the Monty Python film "Monty Python and the Holy Grail." However, there is no definitive answer to this question as it is based on a line from a fictional movie and there is no scientific evidence to support an exact value. Swallows, specifically the African Swallow (Hirundo rustica), have been estimated to fly at speeds between 11 and 12 meters per second (approximately 25 to 27 miles per hour) based on observations and studies. However, these estimates do not account for the weight of the swallow carrying food or eggs, making the question of the airspeed velocity of an unladen swallow a matter of folklore rather than fact. |

Table 16: **Comparison of model responses to the medical-domain question from AlpacaEval2.0.** All models are based on **Mistral** and trained with **10% of UltraMedical-Preference**. This table includes the outputs from our baselines: DPO, ORPO, SimPO, and SSRM. Baselines converge on antibody-level summaries and fail to capture the structured, multi-stage mechanisms emphasized in SSPO.

| Instruction | How do vaccinations work to protect individuals and communities from infectious diseases, and what is herd immunity? |
|---|---|
| **DPO** | Vaccinations work by stimulating the immune system to produce an immune response to a pathogen without causing illness. This immune response includes the production of antibodies that can recognize and destroy the pathogen if the person is exposed to it in the future. Herd immunity refers to the protection of a community or population against the spread of an infectious disease. It occurs when a large percentage of the population is immune to the disease, either through vaccination or prior infection, which reduces the likelihood of the disease spreading within the community. Herd immunity is especially important for protecting individuals who cannot receive vaccines due to medical reasons or those who have weakened immune systems. |
| **ORPO** | Vaccinations work by stimulating the immune system to produce an immune response against specific pathogens without causing the disease. This is achieved through the introduction of antigens, which are components of the pathogen, into the body. The antigens stimulate the production of antibodies, which can recognize and neutralize the pathogen if encountered again. Herd immunity, also known as community immunity, occurs when a significant portion of the population is immune to a particular infectious disease, either through vaccination or prior infection. When a large proportion of the population is immune, the spread of the disease is limited, providing indirect protection to those who are not immune. This is especially important for individuals who cannot be vaccinated due to medical reasons, such as infants, pregnant women, or individuals with weakened immune systems. Overall, vaccinations and herd immunity play a crucial role in protecting individuals and communities from infectious diseases by preventing the transmission and spread of pathogens. |
| **SimPO** | Vaccinations work by stimulating an individual's immune system to produce an immune response against specific pathogens (such as viruses or bacteria) without causing the disease itself. This immune response includes the production of antibodies that can recognize and destroy the pathogen if the person is later exposed to it. Herd immunity, also known as community immunity, refers to a situation where a large proportion of the population becomes immune to a particular infectious disease, either through vaccination or prior infection. When a high percentage of individuals in a community are immune, the spread of the disease is significantly reduced, protecting not only those who are immune but also individuals who are not immune, such as those who are too young or have compromised immune systems. Herd immunity is particularly important for protecting those who cannot be vaccinated due to medical reasons. It also helps prevent outbreaks and epidemics by reducing the number of susceptible individuals in a population. Overall, vaccinations and herd immunity are crucial for controlling and preventing the spread of infectious diseases. |
| **SSRM** | Vaccines are medical interventions that are designed to prevent infectious diseases by stimulating an individual's immune system to produce immunity to a specific pathogen. They contain a weakened or killed version of the pathogen or a small part of it (antigen), which is used to trigger an immune response that produces antibodies and white blood cells that can fight the pathogen. When vaccines are administered to individuals, they provide protection against the targeted disease. However, when a sufficient number of people in a community receive the vaccine, they can create herd immunity. Herd immunity is a situation where a large portion of the population is immune to an infectious disease, which prevents the spread of the disease even if some people in the community are not vaccinated or vaccinated individuals are exposed to the pathogen. This collective immunity occurs because vaccinated individuals cannot transmit the disease to others, and they reduce the pool of susceptible individuals that the pathogen can infect. Consequently, the fewer susceptible individuals in the community, the less the pathogen can spread, which can ultimately lead to the elimination of the disease from the community. Overall, vaccines are an essential tool in public health interventions that can prevent infectious diseases from spreading and promote herd immunity. By increasing vaccination rates, communities can protect themselves and reduce the risk of epidemics and pandemics. |

Table 17: (Continuation of Table 16) **Comparison of model responses to the medical-domain question from AlpacaEval2.0.** All models are based on **Mistral** trained with **10% of UltraMedical-Preference**. This table includes the outputs from SSPO and our strongest baseline: SPA. SSPO provides a highly structured explanation covering vaccine platforms, immune memory, and medical terminology, outperforming all the baselines. For brevity, some contents are omitted.

| Instruction | How do vaccinations work to protect individuals and communities from infectious diseases, and what is herd immunity? |
|---|---|
| KTO | Vaccines keep people from getting sick by making you immune. If enough people get them then diseases go away. Herd immunity is just when the community doesn't get sick anymore. Basically, vaccines are useful because they prevent disease. |
| SPA | Vaccines help protect people by training the immune system to recognize and fight off specific pathogens, such as viruses or bacteria, without causing the actual disease. They do this by prompting the body to produce antibodies, which can quickly respond if the real pathogen is encountered later on. 

 Herd immunity happens when a significant portion of the population becomes immune to a contagious disease, either through vaccination or previous infection. This widespread immunity makes it harder for the disease to spread, thereby offering indirect protection to those who are not immune, such as individuals with certain medical conditions or weakened immune systems who cannot be vaccinated. |
| SSPO | 1. Fundamental Mechanisms: 

 - **Pathogen Recognition and Immune Response**: Vaccines present antigens that mimic or are the actual components of pathogens. Upon administration, these induce an immune reaction including antigen presentation to T-cells, antibody production by B-cells, and the formation of memory cells. This results in the recognition and rapid immune response targeting the specific microbial agents upon subsequent encounters. 

 - The **AHA (antigenic heterogeneity and antigenicity)** of viruses and pathogens necessitates repeated vaccine updates or multiple components within vaccines to cover diverse strains effectively. 

 - **Vaccine Delivery Systems**: Various platforms are used to elicit potent and appropriate immune responses, ranging from live attenuated organisms to protein subunits, nucleic acids encoding viral proteomes, to live vectors (like the chimeric viruses in viral vector vaccines) or even intracellular replicons (like virus-like particles or live attenuated virus replicons in cellular vectors). These approaches tailor the immune stimulus specifically to the target tissue or pathogen lifecycle stage that is crucial for the desired immunity. 

 - **Humoral vs. Cell-Mediated Immunity**: Vaccines generally drive both but emphasize antibody production which can neutralize pathogens extracellularly. Importantly, the establishment of robust T-cell immunity (particularly cytotoxic T lymphocytes for intracellular pathogens or helper T-cell responses that assist in antigen presentation for all pathogens) is crucial in many vaccine designs, especially with certain vaccine modalities and infections. 

 - **Long-term Protection and Memory**: Successful vaccination leads to the establishment of long-lived antibody-producing B-cell populations within memory (or germinal center) centers in secondary lymphoid tissues. These can rapidly proliferate upon reinfection, quickly producing neutralizing antibodies, and are instrumental in the concept of "original antigenic sin." *(omitted)* |

Table 18: **Comparison of model responses to the business-domain question from AlpacaEval2.0.** All models are based on **Mistral** and trained with **10% of DSP Business**. While DPO, ORPO, and SimPO successfully generate structured, list-based answers, their content is generic and superficial. They provide a simple enumeration of common marketing tactics without distinguishing foundational strategies from tactical options, lacking the depth of an expert-level response.

| Instruction | What are the important points for brand promotion and brand marketing? |
|---|---|
| **DPO** | 1. Defining the target audience: Understanding the demographics, needs, and preferences of the target market is crucial for creating effective brand promotions and marketing strategies. 2. Developing a strong brand identity: Establishing a consistent brand image, tone of voice, and message across all communication channels is essential for building brand awareness and recognition. 3. Utilizing various marketing channels: A mix of traditional and digital marketing channels such as social media, email marketing, content marketing, search engine optimization (SEO), pay-per-click (PPC) advertising, and public relations can help reach a wider audience and increase brand exposure. 4. Creating engaging content: Producing high-quality, relevant, and shareable content that resonates with the target audience can help generate buzz and increase brand awareness. 5. Building a community: Engaging with customers and building a community around the brand can help foster loyalty and increase brand advocacy. 6. Measuring and analyzing results: Regularly monitoring and analyzing the effectiveness of brand promotions and marketing strategies can help identify areas for improvement and optimize future campaigns. |
| **ORPO** | 1. Define your target audience: Determine who your ideal customer is, their age, income, interests, and behavior patterns. This information will help you create effective marketing strategies that resonate with your target audience. 2. Create a unique brand identity: Your brand should stand out from the competition. Develop a distinct brand voice, logo, and visual identity that communicates your brand's values and personality. 3. Develop a strong online presence: In today's digital age, having a strong online presence is essential. Build a website, social media profiles, and other online channels that showcase your brand and engage with your target audience. 4. Build brand awareness: Use various marketing channels such as advertising, public relations, and content marketing to increase your brand's visibility and reach. 5. Provide exceptional customer service: Providing excellent customer service is crucial for building brand loyalty and reputation. Ensure that your customers have positive experiences with your brand at every touchpoint. 6. Continuously measure and analyze your marketing efforts: Use analytics tools to measure the effectiveness of your marketing campaigns and make data-driven decisions to optimize your strategies. 7. Stay true to your brand values: Your brand's values should be the foundation of all your marketing and promotional efforts. Ensure that your messaging and marketing strategies align with your brand's mission and purpose. |
| **SimPO** | 1. Know Your Target Audience: Determine your ideal customer and what drives their purchasing decisions. 2. Develop a Strong Brand Identity: Create a clear and consistent brand message that resonates with your audience. 3. Build a Strong Online Presence: Utilize social media, a website, and other online channels to reach your audience and establish your brand. 4. Provide Exceptional Customer Service: Provide excellent customer service to build loyalty and attract repeat business. 5. Leverage Influencer Marketing: Partner with influencers in your industry to reach a wider audience and increase brand awareness. 6. Monitor and Measure Results: Track and analyze your marketing efforts to understand what works and what doesn't, and adjust your strategy accordingly. 7. Offer Value: Provide value to your customers through promotions, discounts, and other incentives to encourage them to engage with your brand. 8. Collaborate with Other Brands: Collaborate with complementary brands to reach new audiences and increase brand visibility. 9. Stay Adaptable: Stay up-to-date on industry trends and adapt your marketing strategy to remain relevant and effective. |

Table 19: (Continuation of Table 18) **Comparison of model responses to the business-domain question from AlpacaEval2.0.** All models are based on **Mistral** trained with **10% of DSP Business**. The baselines here exhibit clear deficiencies: KTO's response is overly simplistic and unhelpful, while SPA's is conceptually vague and fails to provide the actionable, enumerated points requested by the prompt. In contrast, SSPO provides a response that is both comprehensive and practical, delivering a well-organized, actionable guide that successfully avoids the pitfalls of other baselines and demonstrates superior alignment with the user's intent. For brevity, some contents are omitted.

| Instruction | What are the important points for brand promotion and brand marketing? |
|---|---|
| SSRM | Brand promotion and brand marketing are essential components of any successful business strategy. Here are some important points for effective brand promotion and marketing: 1. Define your brand: Understand your brand's mission, vision, values, and unique selling proposition (USP) to communicate a clear and consistent message to your audience. 2. Know your audience: Understand your target audience's demographics, interests, and pain points to tailor your brand messaging and marketing efforts effectively. 3. Build a strong brand identity: Develop a visual and messaging identity that aligns with your brand and resonates with your audience. 4. Utilize multiple marketing channels: Leverage various marketing channels, including social media, email marketing, content marketing, influencer marketing, and paid advertising, to reach your audience where they are. 5. Consistency is key: Maintain a consistent brand message and visual identity across all marketing channels and touchpoints to build trust and recognition with your audience. 6. Engage with your audience: Build a community around your brand by engaging with your audience on social media, responding to customer feedback, and providing excellent customer service. 7. Monitor and measure your results: Use analytics tools to track your marketing efforts' performance and adjust your strategy as needed to optimize your brand's reach and impact. *(omitted)* |
| KTO | Brand promotion and marketing are important because they help people know about your brand. You should try to tell people what your brand is and make sure they like it. Using the internet or talking to customers can help. Overall, the key point is just to make your brand more visible. |
| SPA | Brand promotion and brand marketing are about making sure your brand is noticed, remembered, and appreciated by people. To do this effectively, it's important to focus on visibility and consistency. A brand that is consistently shown to audiences will naturally build recognition over time, which is one of the key elements of strong promotion. Another important point is to use multiple ways of reaching your audience. Different people interact with brands in different environments, so making sure that your brand shows up across a variety of touchpoints will help create familiarity. Whether it's online or offline, what matters most is that your message is delivered in a way that connects with people and stays in their minds. Marketing also plays a role in shaping how customers feel about the brand. If promotion makes people aware, then marketing helps them develop a favorable impression. This means you should think about how to create positive associations and keep reinforcing them. When customers feel good about a brand, they are more likely to engage with it. Ultimately, the most important points for brand promotion and brand marketing can be summarized as visibility, consistency, and recognition. If people know about your brand, see it often, and think positively about it, then your promotional and marketing efforts will be successful. |
| SSPO | 1. Define your brand: Before you start promoting and marketing your brand, it's important to define what it stands for and what sets it apart from competitors. 2. Know your target audience: Knowing your target audience is critical to effective brand promotion and marketing. Understanding their needs, preferences, and pain points will help you create messaging and content that resonates with them. 3. Develop a strong brand identity: Your brand identity includes all the visual and verbal elements that represent your brand. This includes your logo, website design, messaging, and tone of voice. 4. Leverage social media: Social media platforms are powerful tools for brand promotion and marketing. They allow you to reach a wider audience, engage with your target market, and build brand awareness. 5. Create valuable content: Content marketing is a highly effective way to promote and market your brand. Creating valuable, relevant, and informative content will help attract and retain customers. 6. Use influencer marketing: Partnering with influencers in your industry can help you reach a wider audience and build credibility with potential customers. 7. Monitor and analyze your results: Measuring the success of your brand promotion and marketing efforts is important. Regularly tracking and analyzing your results will help you identify what's working and what needs to be improved. *(omitted)* |

