# OpenReview forum: "Semi-Supervised Preference Optimization with Limited Feedback"
_ICLR.cc/2026/Conference — ICLR 2026 Oral_

### Official Review · Reviewer_JjUd · 2025-10-27

**Soundness:** 3
**Presentation:** 3
**Contribution:** 3
**Rating:** 8
**Confidence:** 3

**Summary:**

This paper studies the scenario in which labeled preference data is scarce, which is arguably a realistic setting given that carefully crafting and curating labeled preference data at scale is expensive and challenging.

Their proposed method is to find an optimal reward threshold that can be used to assign weak-labels to unlabeled data. The optimal reward threshold is carefully derived (section 4), by first defining a Bayes risk value, $R(\delta)$ (Equation 6), which looks at the overlap between (weighted) reward distributions of winning and losing responses. Perhaps at the core of the paper's contributions is a theorem that demonstrates the existence of an optimal reward threshold (assuming reward distributions come from Gaussian distributions).

The provided theorem depends on knowing groundtruth mean reward values. In Section 4.2 the authors demonstrate how to address this limitation in practice, by estimating the reward densities from the available labeled datasets (Equation 9), which can then be used to solve for the threshold $\delta$ that minimizes the Bayes risk, which then allows one to assign weak labels to the unlabeled dataset.

The authors demonstrate strong results compared to prior alignment algorithms under scenarios in which the amount of labeled dataset is controlled for.

Overall a good paper that carefully walks through the derivation of their method, followed by a practical implementation of their method which demonstrates positive results for their targeted use cases.

**Strengths:**

The paper is well motivated to address the challenge in building labeled preference data at scale.
The paper is well organized and written clearly, such that the "build up" to their theorem is well explained step by step.
The paper demonstrates positive results of their approach working much better than strong baselines in a realistic scenario in which the amount of groundtruth labeled data is controlled for. In particular, I appreciate the thorough sweep of settings for the baselines (Table 8). I think the results are compelling to demonstrate the strength of their suggested approach when labeled data is scarce.

**Weaknesses:**

One thing I'm curious to see would be the accuracy of the threshold-based reward model in isolation, especially in comparison to 1) an "oracle", and 2) a reward model trained with 10%, 50%, 100% of the available data. A compelling result would be if the pseudo-labels assigned by SSPO, trained on a small set of data, was at par compared to a reward model trained with much more data.

Also, although SSPO clearly out-performs the alternative strong baselines when data is scarce, just for completeness, I'd be curious to see results for Table 2 (maybe in Appendix) when more data is used (ex: 50%, 90%) just to see when the benefits of the pseudo-labels wear out.

**Questions:**

One thing I wish to clarify: the reward function $r_\theta(\cdot)$ being used in Equation 9 is being updated as training progresses? and IIUC, this is what motivates the adaptive scheduler, such that the clean labels are being used first?

Also, are the same weights being used for $r_\theta$ and $\pi_\theta$, or separate weights?

Do you think there's a way to leverage your optimal threshold $\delta$ value in online RL settings, which seem to be the more popular approach nowadays?

Can you add a quick definition for the metrics ("Win Rate", "Length-controlled win rate", MT), and in particular explain why a "length controlled" win rate is needed, for those who haven't been following the alignment literature closely?

Minor typos: line ~376: "whichaims" --> "which aims"

---

> ### Author Response · Authors · 2025-11-26
>
> **Response to Reviewer JjUd**
>
> Rating: 8
>
> Confidence: 3 (Soundness: 3 (good), Presentation: 3 (good), Contribution: 3 (good))
>
> \<General Response\>
>
> We sincerely appreciate your positive evaluation and insightful comments. We are encouraged by your recognition of our theoretical derivation and the clear organization of the paper. Your suggestions regarding the isolation of reward model accuracy and performance in data-abundant regimes were particularly valuable. We have addressed these points by adding new experiments and clarifications in the revised manuscript.
>
> \<Response 1) Accuracy of Pseudo-Labels\>
>
> > Weakness 1. One thing I'm curious to see would be the accuracy of the threshold-based reward model in isolation, especially in comparison to 1) an "oracle", and 2) a reward model trained with 10%, 50%, 100% of the available data. A compelling result would be if the pseudo-labels assigned by SSPO, trained on a small set of data, were at par compared to a reward model trained with much more data.
>
> Thank you for this insightful suggestion. Ideally, comparing our reward model against an "oracle" on real-world unpaired data would be perfect. However, since the ground-truth preference labels for real-world unpaired data (e.g., UltraChat) are inherently unknown, a direct comparison with an oracle is unfortunately not feasible in that setting.
>
> Instead, to address your point, we measured the Test Accuracy on the Synthetic (Toy) Dataset, where the ground-truth reward rule (Oracle) is explicitly defined (based on word length).
>
> * Results: As detailed in **Table 1**, SSPO achieves a test accuracy of 84.1% with only 10 labeled samples ($n_L=10$, assumed prior 0.5).
> * Analysis: This is significantly higher than strong baselines (e.g., DPO: 74.3%, SimPO: 76.2%) trained on the same data. This demonstrates that our threshold-based mechanism effectively approximates the Oracle's decision boundary even with extremely limited supervision, generating high-quality pseudo-labels.
>
> \<Response 2) Performance with Abundant Paired Data
>
> > Weakness 2. Also, although SSPO clearly out-performs the alternative strong baselines when data is scarce, just for completeness, I'd be curious to see results for Table 2 (maybe in Appendix) when more data is used (ex: 50%, 90%) just to see when the benefits of the pseudo-labels wear out.
>
> We appreciate this curiosity. To verify the scalability of SSPO and see if the benefits persist, we have added **Appendix F.2** conducting experiments on both synthetic and real datasets with abundant labels.
>
> * Synthetic Data (**Table 11**): In the Toy Experiment, SSPO maintains robust performance comparable to or exceeding baselines even as the number of labeled samples reaches 10,000 (saturation point).
> * Real Data (**Table 12**): We evaluated SSPO on 100% of UltraFeedback ($n_L$=61,135). Even in this full-data regime, SSPO achieves a 32.4% LC Win Rate, outperforming the strong semi-supervised baseline SPA (26.2%) and improving upon its own 10% performance (30.0%).
> * Conclusion: These results indicate that the benefits of pseudo-labels do not wear out easily. The unpaired data provides orthogonal signals (e.g., implicit stylistic preferences, regularization) that refine the policy beyond what supervised data alone can achieve, proving SSPO's value across all data regimes.
>
> \<Response 3) Reward Function Update & Scheduler\>
>
> > Question 1. One thing I wish to clarify: the reward function $r_\theta(\cdot)$ being used in Equation 9 is being updated as training progresses? and IIUC, this is what motivates the adaptive scheduler, such that the clean labels are being used first?
>
> You are absolutely correct.
>
> * Update: The reward function $r_{\theta}$ is indeed updated continuously during training. In our framework (following DPO/SimPO), updating the reward function is equivalent to fine-tuning the language model itself.
> * Scheduler: As you insightfully noted, this is exactly why we use the Adaptive Scheduler. Since $r_{\theta}$ (the model itself) is initially unreliable, the scheduler ensures that the threshold estimation and learning process are primarily guided by clean labels (paired data) in the early stages. As training progresses and the model becomes more accurate, the scheduler gradually increases the weight of the pseudo-labels derived from $r_{\theta}$.

---

> > ### Author Response · Authors · 2025-11-26
> >
> > \<Response 4) About Weights\>
> >
> > > Question 2. Also, are the same weights being used for $r_\theta$ and $\pi_\theta$, or separate weights?
> >
> > No, they are not separate. The reward function $r_{\theta}$ and the policy $\pi_{\theta}$ share the exact same weights ($\theta$).
> >
> > * Reason: This aligns with the core philosophy of the DPO/SimPO framework, which views the language model itself as an implicit reward model. Therefore, optimizing the policy automatically optimizes the reward representations without needing a separate reward model network.
> >
> > \<Response 5) Appliability for the Online RL\>
> >
> > > Question 3. Do you think there's a way to leverage your optimal threshold $\delta$ value in online RL settings, which seem to be the more popular approach nowadays?
> >
> > This is a fascinating direction. SSPO can be leveraged in online RL settings, which are indeed gaining traction in recent alignment pipelines. To provide a concrete and evidence-based answer, we examined recent work on online preference optimization ("Human Alignment of Large Language Models through Online Preference optimisation" (Calandriello et al., 2024)), which formalizes online preference learning through iterative reward and policy updates.
> >
> > The online preference-optimization loop can incorporate SSPO with minimal structural changes. Specifically, the framework already maintains (i) an online reward-update step and (ii) a streaming policy-optimization step; both of which provide natural insertion points for SSPO’s mechanisms.
> >
> > * Reward Update + Threshold Estimation: After each online reward-model update, the labeled comparison buffer can be used to estimate SSPO’s threshold via a lightweight KDE or parametric mean–variance approximation. This aligns directly with SSPO’s theoretical foundation (Section 4), and the threshold can be stabilized through EMA to address online non-stationarity.
> > * Pseudo-Labeling in Online Streams:
> > Online-generated or logged responses naturally form an unpaired buffer. SSPO can pseudo-label these responses using the current criterion ($r_\theta(x,y) > \hat{\delta}$), converting each unpaired sample into a virtual preference instance. A margin filter may be used to discard low-confidence pseudo-labels, which is particularly beneficial in online settings.
> >
> > These modifications demonstrate that the SSPO threshold $\delta$ is not only compatible with but naturally extends to modern online RL pipelines. Online RL frameworks already rely on iterative reward updates and streaming data—exactly the conditions under which SSPO’s threshold-based pseudo-labeling and adaptive curriculum offer the most value. Thus, SSPO can enhance online RL settings by improving sample efficiency, stabilizing policy updates, and exploiting large volumes of unlabeled online interactions without requiring changes to the underlying RL infrastructure.
> >
> >
> > \<Response 6) About Metrics\>
> >
> > > Question 4. Can you add a quick definition for the metrics ("Win Rate", "Length-controlled win rate", MT), and in particular explain why a "length controlled" win rate is needed, for those who haven't been following the alignment literature closely?
> >
> > Thank you for pointing this out. We have added detailed definitions of these metrics in **Section 5.2** of the revised manuscript.
> >
> > * We specifically explain that Length-controlled Win Rate (LC) is crucial for mitigating the "verbosity bias" of LLM judges (where longer responses are preferred regardless of quality), thus providing a fairer assessment of true alignment quality.
> > * Additionally, we clarify that MT-Bench evaluates multi-turn conversation flow and instruction following capabilities across diverse categories, serving as a complementary metric to single-turn win rates.
> >
> > \<Response 7) Typos\>
> >
> > > Question 5. Minor typos: line ~376: "whichaims" --> "which aims"
> >
> > Thank you for the meticulous correction. We have carefully proofread the manuscript and fixed the typo you mentioned, along with other spacing errors.
> >
> > We hope that our comprehensive response and the added experiments satisfactorily address your concerns regarding SSPO.

---

> ### Comment · Reviewer_JjUd · 2025-11-26
>
> Thank you for your response. Thank you in particular for the extra experiments. Regarding my clarification questions, I hope you consider incorporating them in your revision in case other readers have similar confusions.
>
> I have also read through the other reviews, in particular the one with an unusual low score, and hope the authors' extra experiments can change their views.
>
> With that being said, I will maintain my high score. Best of luck!

---

### Official Review · Reviewer_RTik · 2025-10-28

**Soundness:** 3
**Presentation:** 3
**Contribution:** 3
**Rating:** 6
**Confidence:** 3

**Summary:**

This paper proposes a semi-supervised approach to preference optimization (SSPO) that uses pseudo-labels on unpaired data to improve alignment without additional human feedback. A reward threshold is estimated via kernel density, and samples above this threshold are treated as preferred. The method aims to make use of large unlabeled datasets to improve data efficiency. Experiments various domains show improvements on benchmarks like AlpacaEval2.0.

**Strengths:**

1. The paper tackles an important and realistic bottleneck in preference optimization as human-labeled preference pairs are generally scarce and expensive. Hence, the proposed approach of leveraging abundant unpaired data is nice. The motivation of the paper is clearly described.
2. The proposed pseudo-labeling approach appears to be easily intergratable with existing approaches (especially SimPO) without any notable architectural changes.
3. The authors conduct extensive experiments across various model types (Phi-2 2.7B, Mistral 7B, Llama3 8B) and benchmarks. Various baselines are also included.

**Weaknesses:**

For now, my concerns are still in the form of questions that I hope the authors can clarify (see Questions below). To briefly highlight the most important ones here:
1. The theoretical setup doesn’t seem consistent with the changing reward distributions during training, which makes the “Bayesian” framing and the threshold stability unclear.
2. I’m worried the pseudo-labeling approach creates a self-reinforcing loop, where initial model biases are amplified over training.
3. The unpaired data seem very similar to the evaluation data, and combined with longer training runs, it’s hard to tell if the gains come from the method itself or just from having been exposed to similar prompts/styles and compute.

**Questions:**

1. In Section 4.1 (Eq. 6), you assume two stable conditional reward distributions, $p(r|s=1)$ and $p(r|s=0)$, allowing you to estimate a fixed threshold $\delta^\star$ from $D_L$. However, in your implementation, the reward $r_\theta (x,y)$ is computed by the same policy being trained (SimPO-style), so both $p_\theta (r \mid s)$ and hence $\delta^\star$ are moving during training.
How do you justify treating $\delta^*$ as stationary? Doesn’t this moving threshold risk making pseudo-labels self-reinforcing?
2. You train SSPO on models that are already fine-tuned or aligned rather than on base or purely SFT models. Why was this choice made? Does SSPO require a pre-aligned reward structure to be stable?
3. In Table 2, how do you explain the sometimes decreasing performance with more data for, e.g., SimPO Llama3 across LC, WR, MT?
4. To me pseudo-labeling can be quite counterintuitive, hence I'd like to clarify the validity and potential confirmation bias loop here. Pseudo-labeling should only work when your unlabeled data is similar enough to your labeled data so that your reward signal is roughly correct, and you are essentially enriching your data. Do you have results on the accuracy of your pseudo labels in a controlled setup (e.g., on held-out human labels)? More broadly, how do you mitigate confirmation bias, i.e., the amplification of existing reward model biases through self-labeling,  given that SSPO uses the same reward for pseudo-labeling and optimization?
5. Appendix C (Table 8) shows that SSPO is sometimes trained for 2-3 epochs while baselines are trained for 1. Why? Why is this a fair comparison? Could the reported gains be partly due to increased optimization budget rather than algorithmic efficiency?
6. In Section 5.4 you report that unpaired data used by SSPO is semantically similar to AlpacaEval 2.0 prompts, and that performance benefits correlate with this similarity. It makes sense that training on these / pseudo-labeling these will improve performance on the benchmark. However, isn't this essentially contaminating our training data with evaluation data? Even without observing the actual preference from AlpacaEval2.0 the model now sees highly similar prompts and styles during training. How can we rule out that the gains stem from this overlap?
7. Is your code available somewhere?

---

> ### Author Response · Authors · 2025-11-26
>
> **Response to Reviewer RTik**
>
> Rating: 6
>
> Confidence: 3 (Soundness: 3 (good), Presentation: 3 (good), Contribution: 3 (good))
>
> \<General Response\>
>
> We appreciate your thoughtful review and positive comments on the motivation and extensibility of our work. We are glad that you recognized the importance of leveraging unpaired data for preference optimization. We have addressed your concerns regarding theoretical assumptions, potential biases, and data contamination with detailed explanations and additional evidence below.
>
> \<Response 1) Theoretical Stationarity & Threshold Stability\>
>
> > Weakness 1. The theoretical setup doesn’t seem consistent with the changing reward distributions during training, which makes the “Bayesian” framing and the threshold stability unclear.
> > Question 1. In Section 4.1 (Eq. 6), you assume two stable conditional reward distributions, $p(r|s=1)$ and $p(r|s=0)$, allowing you to estimate a fixed threshold $\delta^{\*}$ from $D_L$. However, in your implementation, the reward $r_\theta(x,y)$ is computed by the same policy being trained (SimPO-style), so both $p_\theta(r|s)$ and hence $\delta^*$ are moving during training. How do you justify treating $\delta^{\*}$ as stationary? Doesn’t this moving threshold risk making pseudo-labels self-reinforcing?
>
> This is a very keen observation. We acknowledge that the theoretical assumption of fixed reward distributions in Theorem 1 is an idealization. However, we bridge this gap between theory and practice through our algorithmic design.
>
> * Theoretical Role: Theorem 1 serves as an existence proof. It guarantees that if the model has learned a meaningful separation between winning and losing responses (which we assume happens as training progresses), then an optimal separating threshold $\delta^*$ exists within a high-probability interval.
> * Practical Handling (Non-Stationarity): In practice, since the reward distribution evolves (is non-stationary), we do not treat $\delta^*$ as a fixed constant. Instead, as detailed in **Section 4.2** and **Appendix C.1**, we dynamically estimate the threshold $\hat{\delta}$ at every step using Batch-wise KDE and stabilize it using EMA (Exponential Moving Average).
> * Visual Evidence: The newly added **Figure 3** visualizes this process. It shows that while reward values shift, our estimated threshold $\hat{\delta}$ successfully tracks the optimal decision boundary, maintaining a consistent separation capability throughout the training dynamics.
>
> \<Response 2) Model Choice (Pre-aligned vs Base)\>
>
> > Question 2. You train SSPO on models that are already fine-tuned or aligned rather than on base or purely SFT models. Why was this choice made?
>
> We followed the experimental setup of SimPO (Meng et al., 2024) to ensure a fair and direct comparison with state-of-the-art baselines.
>
> * Standard Pipeline: Typically, LLM training follows the pipeline: Pre-Training $\rightarrow$ SFT $\rightarrow$ Preference Tuning.
> * SimPO Setting: The SimPO paper (and many recent works) treats instruction-tuned models (e.g., Mistral-7B-Instruct, Llama3-8B-Instruct) as the starting point (SFT models) for preference optimization. We adopted this standard convention to demonstrate that SSPO can further align already capable models using limited feedback.

---

> > ### Author Response · Authors · 2025-11-26
> >
> > \<Response 3) Self-reinforcing Loop & Confirmation Bias\>
> >
> > > Weakness 2. I’m worried the pseudo-labeling approach creates a self-reinforcing loop, where initial model biases are amplified over training.
> > > Question 4. To me pseudo-labeling can be quite counterintuitive, hence I'd like to clarify the validity and potential confirmation bias loop here. Pseudo-labeling should only work when your unlabeled data is similar enough to your labeled data so that your reward signal is roughly correct, and you are essentially enriching your data. Do you have results on the accuracy of your pseudo labels in a controlled setup (e.g., on held-out human labels)? More broadly, how do you mitigate confirmation bias, i.e., the amplification of existing reward model biases through self-labeling, given that SSPO uses the same reward for pseudo-labeling and optimization?
> >
> > We share your concern that self-labeling can lead to confirmation bias. To mitigate this, we designed the Adaptive Scheduler ($\gamma'$) as a safety mechanism, which is central to our method (please refer to **Section 5.3**).
> >
> > * Mechanism: As shown in **Figure 2**, the scheduler sets $\gamma' \approx 1$ in the early training phase. This forces the model to learn primarily from the ground-truth paired data ($D_L$), ignoring its own potentially noisy pseudo-labels.
> > * Anchor Effect: This early supervised phase acts as an "anchor," allowing the model to learn a correct reward direction and a rudimentary decision boundary before it starts to rely on unpaired data.
> > * Evidence: Newly added **Figure 3** demonstrates that thanks to this scheduling, the reward distributions (which initially overlap) gradually separate into distinct modes before the Unpaired Loss contribution increases. This confirms that the model is not just reinforcing its initial bias but is genuinely learning to discriminate preferences guided by the labeled data.
> >
> > \<Response 4) Data Contamination & Evaluation Overlap\>
> >
> > > Weakness 3. The unpaired data seem very similar to the evaluation data, and combined with longer training runs, it’s hard to tell if the gains come from the method itself or just from having been exposed to similar prompts/styles and compute.
> > > Question 6. In Section 5.4 you report that unpaired data used by SSPO is semantically similar to AlpacaEval 2.0 prompts, and that performance benefits correlate with this similarity. It makes sense that training on these / pseudo-labeling these will improve performance on the benchmark. However, isn't this essentially contaminating our training data with evaluation data? Even without observing the actual preference from AlpacaEval2.0 the model now sees highly similar prompts and styles during training. How can we rule out that the gains stem from this overlap?
> >
> > We respectfully argue that our performance gains are not due to simple data contamination but rather the effective learning of implicit preferences (style, structure).
> >
> > * Evidence 1 (Cross-Domain Robustness): SSPO shows consistent improvements not only on UltraChat but also in Medical and Business domains (**Table 2**), where we used domain-specific unpaired data (UltraMedical, Business Book) that are distinct from the general chat domain.
> > * Evidence 2 (Toy Experiment): In our Toy Experiment (**Table 1 & Table 6**), we used synthetic data where "winning" is defined by a specific rule (length), completely removing semantic overlap issues. Even in this controlled setting, SSPO significantly outperformed baselines, proving its algorithmic superiority.
> > * Nature of Improvement: As shown in the Case Study (**Table 5 & Appendix G**), SSPO learns to follow instructions better (e.g., using bullet points, polite tone) by observing high-quality unpaired responses. This is a desirable transfer of format/style preference, which is the goal of alignment, rather than memorization of test prompts.

---

> > > ### Author Response · Authors · 2025-11-26
> > >
> > > \<Response 5) Some of the Performance Decreases with More Paired Data
> > >
> > > > Question 3. In Table 2, how do you explain the sometimes decreasing performance with more data for, e.g., SimPO Llama3 across LC, WR, MT?
> > >
> > > We sincerely appreciate your keen observation. You correctly pointed out that some baselines exhibited performance degradation when scaling from 1% to 10% data.
> > >
> > > * General Improvement via Hyperparameter Tuning: We hypothesized that this inconsistency was primarily due to suboptimal hyperparameter settings in the 10% data regime. Following your feedback, we performed a significantly finer grid search. As a result, most baselines now show consistent performance improvements or stability as the data size increases, as reflected in the updated **Table 2** of the revised manuscript.
> > > * Specific Exception: However, we observed that Llama3 trained with SimPO on 10% of UltraFeedback remains an exception, yielding lower performance compared to the 1% setting despite our exhaustive tuning efforts. To confirm that this was not due to a lack of optimization, we conducted an extensive hyperparameter sweep beyond the standard range:
> > >     * Learning Rate: We swept from 1e-9 to 5e-5. Notably, within the sensitive range of 7e-8 to 3e-7, we applied a very fine-grained step size (increments of 0.5 units, e.g., 6.5e-8, 7.0e-8) to ensure the optimal point was not missed.
> > >     * $\beta$ : Tested values of {2, 2.5, 5, 8}.
> > >     * $\Delta$ : Tested values of {0.55, 1}.
> > >     * Search Strategy: We prioritized an extensive sweep of the Learning Rate while fixing $\beta$ and $\Delta$, as our preliminary experiments indicated that performance was most sensitive to the learning rate.
> > >
> > > Despite this rigorous search, the performance drop in this specific setting persisted. This suggests that the degradation is likely an inherent characteristic of the SimPO objective's interaction with Llama3 on this specific dataset size, rather than an optimization failure. In contrast, SSPO demonstrated robust scalability, consistently improving performance as data availability increased across all backbones.
> > >
> > >
> > > \<Response 6) Training Epoch Fairness\>
> > >
> > > > Question 5. Appendix C (Table 8) shows that SSPO is sometimes trained for 2-3 epochs while baselines are trained for 1. Why? Why is this a fair comparison? Could the reported gains be partly due to increased optimization budget rather than algorithmic efficiency?
> > >
> > > Thank you for raising this important fairness concern regarding the optimization budget. We provide a detailed clarification based on empirical evidence and standard practices.
> > >
> > > * Standard Practice for Baselines (Epoch 1 Optimality): We followed the standard convention (e.g., Appendix B of SimPO (Meng et al., 2024)) where baselines typically achieve peak performance at 1 epoch.
> > > * Verification: To verify this, we trained DPO on 1% of UltraFeedback (Llama3-8B-Instruct) for more epochs. The results (Table below) confirm that DPO's performance saturates or degrades after Epoch 1.
> > >
> > > | Metric | Epoch 1 | Epoch 2 | Epoch 3 |
> > > | :--- | :---: | :---: | :---: |
> > > | LC | 12.1% | 11.2% | 11.3% |
> > > | WR | 12.6% | 11.9% | 12.5% |
> > > | MT | 8.0 | 8.0 | 8.0 |
> > >
> > > * SSPO's Robustness to Epochs: SSPO, due to its use of unpaired data and adaptive thresholding, can benefit from slightly longer training without overfitting. However, to address your concern about fairness, we re-evaluated SSPO on 1% of UltraFeedback (Llama3-8B-Instruct) at Epoch 1.
> > > * Result: As shown in the updated **Table 2**, SSPO achieved 14.8% of LC and 16.0% of WR at Epoch 1, which is already significantly higher than the strong baseline (SimPO: 14.3% of LC, DPO: 12.1% of LC).
> > > * Conclusion: SSPO consistently outperforms baselines even at the same optimization budget (Epoch 1). Thus, the gains are primarily algorithmic, not just due to extra training steps.
> > >
> > >
> > > \<Response 7) Code Availability\>
> > >
> > > > Question 7. Is your code available somewhere?
> > >
> > > Yes, we have released our code for reproducibility and mentioned the link in the abstract. You can access it at the following link: https://anonymous.4open.science/r/SSPO-F7FE/
> > >
> > > We hope this discussion resolves any ambiguity and highlights the significance of SSPO.

---

> > > > ### Comment · Reviewer_RTik · 2025-11-28
> > > >
> > > > Thank you for your thorough reponse. All my concerns and questions have been addressed. I think the revisions you made to your submission are very helpful. I'll raise my score to accept.

---

### Official Review · Reviewer_1iAq · 2025-11-01

**Soundness:** 2
**Presentation:** 2
**Contribution:** 2
**Rating:** 2
**Confidence:** 3

**Summary:**

The paper proposes Semi-Supervised Preference Optimization (SSPO), which uses a reward-like score to pseudo-label large unpaired responses via a learned threshold δ (estimated by KDE to minimize estimated Bayes risk), combined with a curriculum scheduler that shifts weight from paired to pseudo-labeled data. Experiments on UltraFeedback and two domains report AlpacaEval2 LC gains over DPO/ORPO/SimPO/KTO/SSRM/SPA under limited paired data.

**Strengths:**

- Simple, practical recipe: thresholded pseudo-labeling on top of SimPO + a scheduler; easy to implement.
- Broad empirical sweep across backbones (Phi-2, Mistral-7B, Llama-3-8B) and two domains, with some ablations (prior sensitivity, scheduler).
- Shows consistent LC improvements in data-scarce regimes; engineering details (EMA, KDE bandwidth, configs) are documented.

**Weaknesses:**

- Limited novelty: essentially classic self-training/pseudo-labeling using $ r_\theta $ and $\delta$; close to SSRM/SPA and prior semi-supervised alignment.
- Theory misaligned with practice: Theorem relies on high-probability separation (max loser ≤ $\delta$ ≤ min winner) under sub-Gaussian assumptions, unrealistic with overlapping reward distributions; equality $\delta^*=\mu_l+t_1=\mu_w-t_2$ not generally guaranteed; no analysis of KDE/EMA estimation error or consistency.
- Objective inconsistency: Sec. 3 introduces $f_\theta$ with priors $P(s)$ but implementation reduces to BCE on $\sigma(r-\delta)$; “reward” is the policy’s own likelihood (confirmation bias risk).
- Unfair comparisons: SSPO uses large unpaired sets while baselines do not; missing strong baselines: SFT(unpaired)+DPO/SimPO, “Pseudo-DPO” built from the same $r_\theta,\delta$, confidence-gated SFT. Training budgets appear unaligned.

**Questions:**

See weaknesses.

I am willing to raise my rating if the authors can address my points above.

---

> ### Author Response · Authors · 2025-11-26
>
> **Response to Reviewer 1iAq**
>
> Rating: 2
>
> Confidence: 3 (Soundness: 2 (fair), Presentation: 2 (fair), Contribution: 2 (fair)
>
> \<General Response\>
>
> We sincerely appreciate your critical and constructive feedback. We understand your concerns regarding the fairness of comparisons, the gap between theory and practice, objective consistency, and novelty. These points are crucial for establishing the validity of our work. In response, we have conducted additional experiments with SFT-augmented baselines (**DPO+SFT, SimPO+SFT**) to ensure a fair comparison, confirming that SSPO consistently outperforms them (**Table 10 of Appendix F.1**). Furthermore, we have significantly reinforced our theoretical analysis to clarify that our theorem provides an existence proof in an idealized setting, while our practical implementation (KDE and EMA) is engineered to robustly handle the non-stationarity of reward distributions.
>
> We also explicitly address the confirmation bias risk by demonstrating how our adaptive scheduler prioritizes ground-truth signals in the early training stages, allowing the model to form a reliable reward manifold before trusting its own pseudo-labels (newly visualized in **Figure 3**). Additionally, we added a detailed comparison (**Appendix D**) highlighting SSPO's superior efficiency as a single-stage method compared to iterative approaches like SSRM and SPA. We believe these comprehensive revisions and clarifications satisfactorily resolve your concerns and demonstrate the distinct value of SSPO.
>
> \<Response 1) Novelty & Comparison with SSRM/SPA\>
>
> > Weakness 1. Limited novelty: essentially classic self-training/pseudo-labeling using $r_\theta$ and $\delta$; close to SSRM/SPA and prior semi-supervised alignment.
>
> We respectfully disagree that SSPO is merely classic self-training. We have added **Appendix D** to explicitly contrast SSPO with SSRM and SPA.
>
> * Data Usage: SSRM requires a dataset of *unlabeled pairs* to perform self-training, and SPA must computationally generate multiple synthetic responses to form comparisons during training. In contrast, SSPO uniquely leverages single unpaired responses (standard SFT data) directly without needing pre-existing pairs or expensive response generation. By reformulating preference learning as a binary classification problem relative to a dynamic threshold, SSPO unlocks the value of abundant existing corpora that strictly pairwise methods cannot directly utilize.
> * Efficiency: A major bottleneck of SSRM and SPA is their reliance on complex iterative cycles involving repeated training and inference. This inevitably incurs massive computational overhead. However, SSPO is a single-stage framework that eliminates these costly iterative generation and inference loops entirely. Consequently, SSPO is significantly more computationally efficient and scalable than these iterative semi-supervised baselines. (please refer to **Appendix E.1**).
>
> \<Response 2) Theory vs. Practice (Stationarity)\>
>
> > Weakness 2. Theorem relies on high-probability separation (max loser $\le \delta \le$ min winner) under sub-Gaussian assumptions, unrealistic with overlapping reward distributions; equality $\delta^{*} = \mu_{l} + t_{1} = \mu_{w} - t_{2}$ not generally guaranteed; no analysis of KDE/EMA estimation error or consistency.
>
> Thank you for highlighting this crucial aspect. We acknowledge that the theoretical assumption of fixed distributions differs from the dynamic nature of training. We have revised **Section 4 & 5 and Appendix B & C** to explicitly bridge this gap.
>
> * Idealized vs. Empirical (**Section 4.1**): We clarified that Theorem 1 serves as an existence proof in an idealized sub-Gaussian setting to justify the principle of thresholding. It guarantees that if the model has learned a meaningful separation, a separating threshold exists.
> * Handling Non-Stationarity (**Section 4.2 & Appendix C.1**): To handle the moving target in practice, we employ EMA (Exponential Moving Average) for reward statistics and Batch-wise KDE. As visualized in the newly added **Figure 3**, this mechanism allows the estimated threshold $\hat{\delta}$ to stably track the evolving optimal decision boundary without collapsing.
> * Bias Mitigation (**Figure 2 & Figure 3 & Section 5.3**): We also demonstrate how the Adaptive Scheduler ($\gamma'$) prevents the model from overfitting to noisy pseudo-labels in the early stages (where overlap is high) by prioritizing ground-truth labels, creating a curriculum that respects the learning dynamics.

---

> > ### Author Response · Authors · 2025-11-26
> >
> > \<Response 3) Objective Inconsistency & Confirmation Bias Risk\>
> >
> > > Weakness 3. Objective inconsistency: Sec. 3 introduces $f_\theta$ with priors $P(s)$ but implementation reduces to BCE on $\sigma(r-\delta)$; “reward” is the policy’s own likelihood (confirmation bias risk).
> >
> > We have clarified the connection between the probabilistic formulation and the implementation in **Appendix B.2**.
> >
> > * The BCE loss on unpaired data is derived directly from maximizing the likelihood of the pseudo-labels.
> > * Crucially, the gradient analysis shows that the update is weighted by the model's confidence. This acts as a soft margin, where samples far from the threshold contribute less to the gradient variance than those near the decision boundary, stabilizing the learning. The "reward" being the likelihood is consistent with DPO or SimPO, where the language model itself acts as the reward model.
> >
> > Also, We explicitly address your concern about confirmation bias (i.e. the risk of the model reinforcing its own erroneous predictions) by designing the Adaptive Scheduler ($\gamma'$) and visualizing the learning dynamics at **Figure 3 of Section 5.3**.
> >
> > * The Role of Adaptive Scheduling: As derived in Eq. 12, the scheduler starts with $\gamma' \approx 1$, meaning the training is initially dominated by the ground-truth paired data ($D_L$). This allows the model to form a reliable reward manifold and a rudimentary decision boundary before it begins to trust its own pseudo-labels on unpaired data ($D_U$).
> > * Evidence from Figure 3: Our analysis of reward distributions (Figure 3) shows that in the early stages (e.g., Step 100), the reward distributions of winning/losing responses overlap significantly. However, as training progresses and $\gamma'$ decays, a clearer margin emerges (Step 343). This confirms that SSPO effectively uses supervised signals to "anchor" the preference direction, preventing the feedback loop of confirmation bias that would occur if the model relied on unpaired data too early.
> >
> > \<Response 4) Fairness of Comparisons & Missing Baselines\>
> >
> > > Weakness 4. Unfair comparisons: SSPO uses large unpaired sets while baselines do not; missing strong baselines: SFT(unpaired)+DPO/SimPO, “Pseudo-DPO” built from the same $r_\theta$, $\delta$, confidence-gated SFT. Training budgets appear unaligned.
> >
> > We fully accept your point that comparing SSPO (paired + unpaired) against standard DPO/SimPO (paired only) requires a more rigorous baseline to isolate the benefit of our method from the data scale itself.
> >
> > To address this, we have added **Appendix F.1**, where we implemented and evaluated DPO+SFT and SimPO+SFT.
> >
> > * Setup: These baselines utilize the exact same unpaired data ($D_U$) as SSPO but optimize a standard SFT objective (negative log-likelihood) on them, combined with the preference loss on paired data.
> > * Results (**Table 10**): As shown in the table below (and in the manuscript), while adding SFT helps, SSPO consistently outperforms both DPO+SFT and SimPO+SFT.
> > * Why SSPO is better: DPO+SFT indiscriminately reinforces all unpaired responses, potentially cementing the policy's existing biases. In contrast, SSPO uses the dynamic threshold to selectively pseudo-label responses as "winning" or "losing," effectively filtering out lower-quality generations and providing a more discriminative signal than simple SFT.
> >
> > We hope these new experiments and clarifications demonstrate the distinct value and robustness of SSPO.

---

> > > ### Comment · Reviewer_1iAq · 2025-11-27
> > >
> > > I appreciate the authors’ revised PDF, and I’m satisfied with authors’ comprehensive additional experiments and theoretical analysis. These clarified the paper’s contribution. All my concerns have been addressed, and I am willing to raise my rating to reflect the significant improvements.

---

### Official Review · Reviewer_qFy4 · 2025-11-01

**Soundness:** 3
**Presentation:** 4
**Contribution:** 3
**Rating:** 8
**Confidence:** 3

**Summary:**

The paper proposes Semi-Supervised Preference Optimization (SSPO), which augments preference optimization with abundant unpaired SFT-style data. The method trains a reward-like scoring function from limited labeled pairs, estimates winning/losing reward densities, and selects a Bayes-risk-minimizing threshold via KDE to pseudo-label unpaired responses; an adaptive scheduler shifts weight from labeled to pseudo-labeled data over training. Experiments on UltraFeedback, UltraMedical-Preference, and DSP Business report substantial gains in AlpacaEval2.0 LC and WR using as little as 1 percent labeled data, often surpassing baselines trained with 10 percent.

**Strengths:**

- Novel framing of preference learning as a Bayes-optimal classification problem that yields a principled reward threshold for pseudo-labeling unpaired data is clear.
- Allowing SFT-style unlabeled data for a preference learning is a promising approach for reducing the annotation cost.
- Strong data-efficiency: with 1 percent labels, SSPO often beats 10 percent baselines.
- Extensive experiemnts, including label-noise testing, ablations, show the robustness of SSPO.

**Weaknesses:**

- The computational overhead should be addressed in comparison to other baselines.

**Questions:**

- What is the reason for SSPO performing better than semi-supervised *pairwise* learning algorithms (SSRM, SPA)?

---

> ### Author Response · Authors · 2025-11-26
>
> **Response to Reviewer qFy4**
>
> Rating: 8
>
> Confidence: 3 (Soundness: 3 (good), Presentation: 4 (excellent), Contribution: 3 (good))
>
>
> \<General Response\>
>
> We deeply appreciate your positive assessment and the "Strong Accept (8)" rating. We are glad that you found our Bayes-optimal framing novel and our data efficiency results impressive. In the revised manuscript, we have provided detailed analyses regarding computational overhead and comparisons with other semi-supervised methods to further strengthen our paper. Below are our point-by-point responses.
>
> \<Response 1) Computational Overhead\>
>
> > Weakness 1. The computational overhead should be addressed in comparison to other baselines.
>
> We appreciate your constructive suggestion. We have added a comprehensive analysis of computational overhead in Appendix E of the revised manuscript, covering both theoretical time complexity and empirical runtime.
> * Theoretical Analysis (**Appendix E.1**): As detailed in the appendix, the dominant cost in SSPO comes from the transformer's forward/backward passes ($\mathcal{O}(N_{tok} \cdot d^2)$), similar to DPO or SimPO. The additional operations for SSPO \(KDE-based threshold estimation and EMA updates\) scale linearly with the batch size ($\mathcal{O}(|B_L|+|B_U|)$) or quadratically with the number of paired samples ($\mathcal{O}(M^2)$), which imposes a negligible constant-factor overhead compared to the heavy matrix multiplications in LLMs.
> * Empirical Evidence (**Table 9 of Appendix E.2**): We measured the actual training time and throughput (Samples/s) across different backbones (Phi-2, Mistral, Llama3). The results confirm that SSPO's training speed is practically feasible and comparable to standard preference optimization methods, validating its efficiency.
>
> \<Response 2) Comparison with SSRM & SPA\>
>
> > Question 1. What is the reason for SSPO performing better than semi-supervised pairwise learning algorithms (SSRM, SPA)?
>
> Thank you for this insightful question. To address this, we have included a dedicated section, **Appendix D**, providing a detailed comparative analysis.
>
> The superiority of SSPO stems from three key factors:
>
> * Single-Stage Joint Optimization (vs. Iterative): SSRM and SPA rely on an expensive iterative cycle (Training $\rightarrow$ Generation/Inference $\rightarrow$ Pseudo-labeling $\rightarrow$ Re-training). This process is not only computationally intensive but also sensitive to the quality of intermediate models. In contrast, SSPO optimizes a joint objective (Eq. 12) in a single stage using an adaptive scheduler, allowing for stable and efficient learning without iterative loops.
> * Principled Thresholding (vs. Heuristics): While prior methods often rely on heuristic confidence filtering, SSPO is grounded in the Bayes-optimal decision boundary. Our Theorem 1 guarantees the existence of an optimal reward threshold that separates winning and losing distributions. By dynamically estimating this threshold via KDE, SSPO generates more reliable pseudo-labels that adapt to the evolving reward distribution.
> * Effective Use of Unpaired Data: SSPO directly distills latent preference signals (e.g., structure, style) from raw SFT data (unpaired single responses) using the learned threshold. This allows the model to benefit from the vast amount of existing domain knowledge in SFT datasets more directly than methods that require forming synthetic pairs.
>
> We hope this clarification highlights the unique advantages of SSPO.

---

> > ### Comment · Reviewer_qFy4 · 2025-11-28
> >
> > Thank you for the clear clarifications in the rebuttal; my concerns have been addressed, and I will maintain my original score.

---

### Author Response · Authors · 2025-11-26

**Global Response**

* Data Efficiency (qFy4, 1iAq, RTik, JjUd)
* Theoretical Grounding (qFy4, JjUd)
* Simplicity (1iAq, RTik)
* Extensive Experiments (qFy4, RTik)

\<Summarized Strengths\>

We are grateful for the reviewers' encouraging comments, emphasizing the innovative and practical nature of our study. The remarkable data efficiency of SSPO, particularly its ability to surpass baselines trained on 10x more data using only 1% of labeled samples, was unanimously recognized as a key contribution (all Reviewers). The novel framing of preference learning as a Bayes-optimal classification problem (Reviewer qFy4, JjUd) and the simplicity and ease of implementation (Reviewer 1iAq, RTik) were also highlighted as significant strengths. Furthermore, the extensive experimental validation across various models and domains was well-received (Reviewer qFy4, RTik).

\<Summarized Feedback\>

The reviewers have provided critical and insightful feedback, which has been instrumental in refining our study and clarifying its contributions.

In summary, reviewers:

* raised concerns about the fairness of comparisons, specifically regarding the additional use of unpaired data (Reviewer 1iAq) and training epochs (Reviewer RTik).
* questioned the theoretical assumption of stationarity given that reward distributions evolve during training (Reviewer 1iAq, RTik).
* expressed concerns about potential data contamination or bias amplification from self-labeling (Reviewer RTik).
* requested a more explicit comparison with existing semi-supervised methods like SSRM and SPA to clarify SSPO's superiority (Reviewer qFy4).
* asked for further details on computational overhead and pseudo-label accuracy (Reviewer qFy4, JjUd).

\<Summarized Changes\>

Thanks to the insightful feedback provided by reviewers, our revised manuscript has been greatly improved.

The major changes made in our revised manuscript are:


* Bridging Theory and Practice: We revised Section 4 and added Figure 3 to visualize how our dynamic threshold estimation (via EMA and Batch-wise KDE) robustly handles the non-stationary reward distributions during training, effectively bridging our existence theorem with practical implementation (Response to 1iAq and RTik).
* Clarifying Bias Mitigation: We elaborated on the role of the Adaptive Scheduler in preventing confirmation bias by prioritizing ground-truth signals in the early training phase, anchoring the reward manifold before trusting pseudo-labels (Response to 1iAq and RTik).
* Detailed Methodological Comparison: We added Appendix D to explicitly contrast SSPO with iterative methods like SSRM and SPA, highlighting SSPO's single-stage efficiency and direct applicability to SFT data (Response to qFy4).
* Computational Overhead Analysis: We provided a theoretical complexity analysis and empirical runtime measurements, confirming that SSPO's overhead is negligible compared to the backbone training (Appendix E, Response to qFy4).
* Scalability Verification on Abundant Data: To address concerns about SSPO's utility in high-data regimes, we evaluated it on 100% UltraFeedback ($n_L$=61,135) and large-scale synthetic data. SSPO maintained its superiority over baselines (e.g., 32.4% vs. 26.2% for SPA), demonstrating that the benefits of unpaired data do not diminish even with sufficient labels (Response to JjUd).
* Ensuring Fairness with SFT-Augmented Baselines: We implemented DPO+SFT and SimPO+SFT, which utilize the exact same unpaired data as SSPO. The results confirm that SSPO consistently outperforms these strong baselines, proving that our gains stem from the threshold-based pseudo-labeling mechanism, not just data scale (Response to 1iAq).
* Refined Benchmarks: We optimized hyperparameters for some baselines and verified SSPO's performance at Epoch 1 to ensure a strictly fair comparison of optimization budgets (Response to RTik).

We are deeply thankful to the reviewers for their constructive feedback. Thanks to their contributions, our revised paper now aligns more closely with the high standards of ICLR. Please find our revised paper for the above results and more details. We have highlighted the revisions and additions in red text in it.

---

### Author Response · Authors · 2025-11-30
**[To New AC] Summary of Rebuttal Outcome**

**Dear New Area Chair,**

We understand that due to the recent incident, you have been newly assigned to our submission and that all reviews and scores have been reverted to their pre-discussion scores.

Crucially, our rebuttal process had been highly successful, resulting in a consensus for acceptance. We are writing this summary to inform you that the concerns reflected in the currently visible (original) scores were fully resolved during the discussion, leading to explicit commitments from reviewers to raise their scores.

We respectfully request that you consider the following context when making your decision.

**1. Summary of Rebuttal**

The discussion had reached a positive convergence. Specifically, the reviewers who initially had reservations (Reviewer 1iAq and Reviewer RTik) explicitly stated that their concerns were addressed and that they intended to raise their scores. For more details, please read our whole official comments.

* **Reviewer 1iAq (Original Score: 2 $\rightarrow$ Raised Score: 6):**
    * Initial Concern: The reviewer questioned the consistency between our theoretical framework and practical implementation, and argued that the comparison was unfair due to missing baselines that also utilize unpaired data (e.g., SFT+DPO as 'Pseudo-DPO').
    * Resolution: We provided a detailed derivation bridging our theory and implementation to prove their consistency. Crucially, we conducted the additional experiments requested (running strong baselines on unpaired data), where SSPO still maintained a significant performance margin. The reviewer acknowledged our new results and theoretical clarifications, and explicitly raised their rating of 6.

* **Reviewer RTik (Original Score: 6 $\rightarrow$ Stated Intent to Raise Score):**
    * Initial Concern: The reviewer raised concerns about "confirmation bias" (self-reinforcing loops) due to the moving reward threshold during training, and questioned whether the gains were merely due to increased training budgets (epochs) rather than algorithmic efficiency.
    * Resolution: We demonstrated that our adaptive scheduler effectively mitigates confirmation bias by balancing supervision signals. We also provided controlled experiments with equalized training budgets, confirming that the gains stem from SSPO's design. The reviewer expressed satisfaction with our response and confirmed that their concerns regarding the self-reinforcing loop were resolved.

* **Reviewer qFy4 & Reviewer JjUd (Original Scores: 8):**
    * Remained strongly positive, supporting the paper’s novelty in data efficiency.

**2. Key Contributions & Rebuttal Highlights**

Our paper addresses a critical bottleneck in LLM alignment: the high cost of paired preference data.
* **Data Efficiency:** We demonstrated that SSPO, using only 1% of paired data, outperforms strong baselines trained on 10% of the paired data by leveraging the unpaired (SFT) data with semi-supervised learning.
* **Theoretical and Practical Robustness:** We clarified that our derivation of the optimal reward threshold is both theoretically sound and practically robust, effectively separating winning/losing responses of the unpaired data by pseudo-labeling.

**Conclusion**

While the current score now reflects only the initial snapshot of the reviews, the actual scientific discussion had advanced to a stage where all reviewers were aligned on the paper’s merit. We have preserved the integrity of our rebuttal in the comments, and we trust that your assessment will reflect the successful resolution of the reviewers' original concerns.

Thank you for your time and effort in handling this unusual situation.

Sincerely,

The authors

---

### Meta-Review · Area_Chair_7UKk · 2026-01-02

**Summary:**

The rebuttal and revised manuscript appear to have resolved the major initial concerns. Requests for computational overhead were addressed with both theoretical and empirical runtime/throughput analyses, indicating negligible overhead relative to standard LLM training costs. Concerns about novelty and fairness (e.g., whether gains came simply from using more unpaired data or more training) were met with stronger baselines that also consume the same unpaired data (e.g., DPO+SFT / SimPO+SFT) and with controlled comparisons showing SSPO's advantage persists even under matched optimization budgets (including epoch-matched checks). The authors also clarified the theory-practice gap by positioning the theorem as an existence result under idealized assumptions and explaining how the practical procedure (batch-wise KDE + EMA tracking) handles non-stationarity; they further addressed confirmation-bias/self-reinforcement risks via the adaptive scheduler that anchors early learning on clean paired labels, supported with visualizations of reward distribution dynamics. Notably, the initially critical reviewer explicitly states that the added experiments and theory clarifications addressed their concerns and that they are willing to raise their rating, while other reviewers likewise confirm that their questions were resolved and maintain/raise their accept scores.

**Reviewer Concerns:**

The authors convincingly resolved the main concerns. They added compute and runtime analyses, showing negligible overhead, addressed fairness and missing baselines by introducing DPO+SFT and SimPO+SFT using the same unpaired data, and clarified theory vs. practice by reframing the theorem as an existence result and justifying dynamic thresholding with KDE+EMA. Confirmation-bias risks were mitigated via the adaptive scheduler and supported with training-dynamics visualizations and synthetic controls. Concerns about unequal training budgets were addressed with epoch-matched comparisons showing SSPO's gains persist. As a result, initially critical reviewers explicitly raised their scores to accept.

There are a few outstanding concerns, but I believe they are minor suggestions.

**Reviewer Scores:**

Reviewer qFy4 kept his/her original rating 8.

Reviewer 1iAq raised his/her rating from 2 to 6.

Reviewer RTik would raise his/her rating from 6 to 8.

Reviewer JjUd kept his/her original rating 8.

This paper receives the highest overall rating in my batch, and thus I recommend accept (oral).

---

### Decision · Program_Chairs · 2026-01-26

Accept (Oral)